# From Raw Corpora to Domain Benchmarks: Automated Evaluation of LLM Domain Expertise

## Abstract

Accurate domain-specific benchmarking of LLMs is essential, specifically in domains with direct implications for humans, such as law, healthcare, and education. However, existing benchmarks are documented to be contaminated and are based on multiple-choice questions, which suffer from inherent biases. To measure domain-specific knowledge in LLMs, we present a deterministic pipeline that transforms raw domain corpora into completion-style benchmarks without relying on other LLMs or costly human annotation. Our approach first extracts domain-specific keywords and related target vocabulary from an input corpus. It then constructs prompt-target pairs where domain-specific words serve as prediction targets. By measuring LLMs' ability to complete these prompts, we provide a direct assessment of domain knowledge at low computational cost. Our pipeline avoids benchmark contamination, enables automated updates with new domain data, and facilitates fair comparisons between base and instruction-tuned (chat) models. We validate our approach by showing that model performances on our benchmark significantly correlate with those on an expert-curated benchmark. We then demonstrate how our benchmark provides insights into knowledge acquisition in domain-adaptive, continual, and general pretraining. Finally, we examine the effects of instruction fine-tuning by comparing base and chat models within our unified evaluation framework. In conclusion, our pipeline enables scalable, domain-specific, LLM-independent, and unbiased evaluation of both base and chat models.

## 1 Introduction

The rapid proliferation of large language models (LLMs) has produced a vast and growing landscape of both general-purpose and specialized models, spanning diverse architectures, scales, and training recipes (Chiang et al., 2024). This abundance raises a fundamental question for practitioners: *given a specific domain of interest, which model is the most knowledgeable?* Whether selecting a model for a specialized application or choosing a base model from which to start domain adaptation (Han et al., 2023; Colombo et al., 2024; Singhal et al., 2025; Liu et al., 2023), quantifying domain expertise reliably is essential, since models with stronger domain knowledge adapt more efficiently (Gururangan et al., 2020). Despite this need, no scalable, standardized framework exists to rank models by their domain-specific knowledge.

Existing approaches measure proxies of domain knowledge rather than domain knowledge itself. *Perplexity*, the most common metric for comparing language models, aggregates prediction quality over all tokens, domain-relevant and domain-irrelevant alike, making it impossible to disentangle genuine domain knowledge from general linguistic fluency (Öncel et al., 2024; Xu et al., 2024; Meister & Cotterell, 2021). *Multiple-choice question (MCQ) benchmarks* such as MMLU (Hendrycks et al., 2020) suffer from well-documented biases: Gupta et al. (2024) show that simply reordering answer choices can substantially alter model accuracy, while Chandak et al. (2025) demonstrate that popular MCQ benchmarks can often be answered without even seeing the question, calling into question whether they measure knowledge at all. MCQ formats further disadvantage base models, which lack instruction-following capabilities and are sensitive to few-shot formatting (Brown et al., 2020; Alzahrani et al., 2024). Beyond format-level issues, *benchmark contamination*, where models are evaluated on data they were trained on, undermines the validity of static benchmarks and is difficult to detect post hoc (Zhou et al., 2023). Finally, existing domain-specific benchmarks such as MedQA (Jin et al., 2021)

Figure 1: Issues with existing domain-specific benchmarks: Perplexity aggregates predictions over all tokens (including domain-irrelevant ones); performance on multiple choice questions depend on the order of options; many benchmarks are already incorporated in the training sets of LLMs; and manual creation is too expensive.

and LegalQA (Louis et al., 2024) organize around broad categories rather than fine-grained subdomains, and rely on formats (MCQs, long-form answers) that make fair comparison across model types difficult.

In this work, we present a deterministic pipeline that transforms any raw domain corpus into a completion-style benchmark with prompt–target pairs, without requiring manual curation or reliance on other LLMs. By measuring a model's ability to predict domain-specific target terms given contextual prompts, we directly assess domain knowledge through the same next-token prediction objective on which all LLMs, both base and instruction-tuned, are pretrained. We evaluate models by computing the rank of the correct target token in the model's output distribution, providing a soft and interpretable metric of domain expertise.

A conceptual overview of our approach is shown in Figure 2. Starting from raw domain text, in our case, academic papers and their abstracts (Sec. 2.1), our pipeline first identifies domain concepts by extracting and refining keywords through extensive pre- and post-processing (Sec. 2.2). Each sentence in the full papers is matched with relevant keywords to focus the evaluation on domain-relevant content (Sec. 2.3). Then, we construct a *target vocabulary* of domain-specific phrases from the matched sentences (Sec. 2.4), and compile hundreds of prompt–target pairs per keyword (Sec. 2.5). For example, for the `cs.AI` domain, example keywords include "machine learning" and "reinforcement learning"; the target vocabulary for reinforcement learning includes "Policy", "rewards", and "replay"; and finally an example prompt-target pair is "Prior attempts at improving data efficiency in reinforcement learning, involved the use of an Experience"-"replay". Models are evaluated based on their ability to predict these targets from given prompts (Sec. 2.6).

Empirically, we first highlight the pitfalls of MCQ benchmarks by showing that model rankings shift substantially under option reordering and prompting variations (Sec. 3.1). We then validate our pipeline against a manually curated expert benchmark derived from the textbook *Understanding Deep Learning* (Prince, 2023), obtaining near-perfect correlation ($r=0.99$, $p<0.001$) between model outputs computed on the expert benchmark and the benchmark generated by our pipeline. In controlled domain adaptation experiments (Sec. 4.1), models trained on domains semantically close to a target domain consistently achieve better predictions, while the alternative perplexity and attribution rate (Geva et al., 2023) metrics fail to provide reliable signals. We further demonstrate that our benchmark tracks knowledge acquisition throughout general pretraining of OLMo-2 (OLMo et al., 2024) and continual pretraining of Llama2-7B (Sec. 4.2–4.3), capturing nuanced learning dynamics that MCQ evaluations and perplexity miss entirely. Finally, by evaluating six model families in both base and instruction-tuned variants across four domains—CS.AI, Physics, Biology, and Economics (Sec. 4.4)—we reveal that instruction tuning generally degrades domain knowledge, an "alignment tax" that is highly heterogeneous across architectures and domains.

Our framework offers a fundamentally different approach to domain-specific evaluation through two core innovations. First, **automation at scale**: the pipeline is fully deterministic and requires no human annotation or LLM assistance, meaning practitioners can generate domain-specific benchmarks on demand from any raw

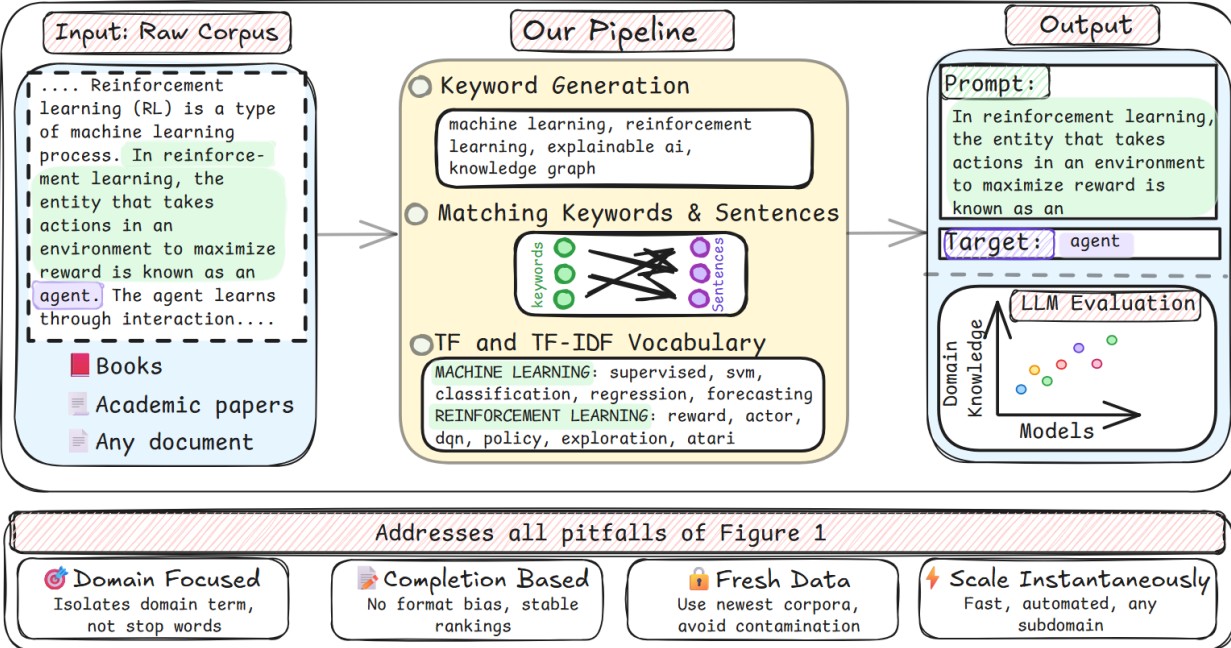

Figure 2: A conceptual overview of our pipeline for generating a completion-based benchmark from a raw domain corpus. We extract and refine keywords from the input corpus. We then match each sentence with relevant keywords to focus evaluation on domain-relevant content. For every keyword, we construct a target vocabulary by collecting domain-specific phrases from the matched sentences; these phrases serve as prediction targets. The domain expertise is quantified by how well the models predict the targets from given prompts.

text corpus. This enables evaluation at arbitrary levels of granularity, from broad disciplines down to specific research areas within a field. Second, **contamination resistance by design**: because benchmarks can be regenerated from fresh or held-out corpora at any time, the pipeline eliminates benchmark contamination as a concern by construction rather than requiring unreliable post-hoc detection.

## 2 Methods

This section outlines our benchmark construction and model evaluation pipeline (see Figure 2 for an overview). While we present our methodology using academic papers spanning diverse scientific domains, it is broadly applicable to any domain-specific corpus, e.g. law, healthcare, and education. We initially developed and refined the pipeline on the `CS.AI` domain in arXiv, leveraging our domain expertise to validate design choices and parameter settings. We then freeze the pipeline and apply it to three additional domains (detailed in Sec. 2.1). The consistent results across all domains demonstrate the robustness and generalizability of our approach beyond the domain in which it was developed.

### 2.1 Data Curation

Our pipeline requires two inputs per document: a short summary (e.g., an abstract) for keyword extraction, and the full text for sentence-level benchmark construction. We instantiate this using RedPajama-Data-1T (Computer, 2023), which contains the full texts of 1.56 million arXiv papers with a token count of 28 billion, spanning multiple STEM disciplines. The dataset consists of preprocessed LaTeX source files with removed preambles, comments, macros, and bibliographies, accessed through the HuggingFace library (Wolf et al., 2019). To obtain structured metadata, we linked these documents to the arXiv Dataset available on Kaggle, which provides titles, authors, categories, submission dates, abstracts, and DOI information. An example data entry is presented in Appendix A.4. Both datasets are publicly available for academic research use.

**Studied domains**  We select four diverse evaluation domains that span different scientific fields: Computer Science – Artificial Intelligence (`CS.AI`), Physics and Society (`Physics (Soc-Ph)`), Quantitative Biology – Populations and Evolution (`Q-Bio.PE`), and General Economics (`Econ-GN`). Our validation covers domains with varying corpus sizes, vocabularies, and degrees of overlap with typical pretraining data.

## 2.2  Keyword Generation

Next, we describe a four-stage procedure for extracting domain-specific keywords. This pipeline operates effectively on a subset of the data, eliminating the need to process the full corpus. Accordingly, we use only the abstracts from Kaggle to extract relevant keywords:

- First, we apply a custom preprocessing routine to standardize the abstracts, including case normalization, removal of bracketed content, consistent handling of contractions, and tokenization that preserves hyphenated terms to retain technical precision.

- Second, we construct n-grams (2–7 tokens) using Gensim's (version 4.3.3) Phrases model (Rehurek & Sojka, 2010; Mikolov et al., 2013) with default parameters. Then we remove *(i)* stopwords (e.g., "the", "is"), *(ii)* generic academic vocabulary (e.g., "paper", "framework"), *(iii)* function words (e.g., "between", "despite"), and *(iv)* quantitative expressions (e.g., "more", "significant").

- Third, we apply adaptive length-based filtering to retain a proportionally balanced subset of n-grams, adjusting length ratios across domains to target 300 high-quality keywords per domain. We note that more keywords enable finer domain coverage but require greater computational resources, a trade-off practitioners can adjust based on their specific objectives. Our experiments showed consistent results with 150 and 450 keywords, demonstrating robustness to this parameter choice.

- Finally, we reduce redundancy by computing pairwise cosine similarities between keywords using the `all-MiniLM-L6-v2` (Wang et al., 2020) from the Sentence Transformer library (Reimers & Gurevych, 2019) and merging semantically similar entries with similarity scores above 0.85. This threshold balances two competing objectives: ensuring sufficiently strong semantic connections to eliminate true redundancy while avoiding overly restrictive matching that would merge distinct but related concepts. Lower thresholds (e.g., 0.7) risk conflating semantically distinct keywords, while higher thresholds (e.g., 0.95) may retain near-duplicate entries that provide minimal additional coverage.

Please see Sec. A.1 for details of each step. Table 4 lists ten representative keywords generated by our pipeline.

## 2.3  Matching Keywords and Sentences

We convert documents from RedPajama into individual sentences using spaCy's `en_core_web_sm` model (Honnibal et al., 2020) and embed both sentences and keywords from Section 2.2. Using cosine similarity with a threshold of 0.5, we extract sentences semantically related to each keyword. We empirically observed that lower values include loosely related or irrelevant sentences that dilute domain specificity, while higher values overly restrict the set of sentences and reduce benchmark diversity. Selected sentences undergo systematic cleaning, including LaTeX formatting conversion, citation removal, and character validation to obtain clean sentence mappings for each domain keyword (see Appendix A.2 for implementation details).

## 2.4  TF and TF-IDF Target Vocabulary

Next, we extract domain-specific terms from the matched sentences, which later serve as prediction targets. Our pipeline has two variants that relate terms and keywords through term frequency (TF) and term frequency–inverse document frequency (TF-IDF).

**TF target vocabulary**  For a given keyword, we combine its matched sentences into a keyword-specific text corpus and compute term and document frequencies. To ensure targets represent domain-specific rather than generic terms, we exclude words that appear in more than 80 percent of all keyword corpora (e.g., the word *learning* for the `CS.AI` domain), as well as stop words and words with one or two characters. This yields an initial term-document matrix of raw TF counts.

Table 1: Example keywords with their TF and TF-IDF target vocabularies for the `CS.AI` domain.

| KEYWORD | TF TARGETS | TF-IDF TARGETS |
|---|---|---|
| machine learning | Backpropagation, Support, Vector, SVM, Random, Machines, Supervised, Bayes, Regression, Forest | Logistic, Naive, AutoML, Boosting, SVMs, kNN, XGBoost, AdaBoost, Scikit, Perceptron |
| reinforcement learning | Replay, Policy, rewards, actor, imitation, MDP, transition, cumulative, critic, Optimization | imitation, critic, Actor, Inverse, MDPs, shaping, replay, bandit, Bellman, Proximal |

**TF-IDF target vocabulary**  For each keyword, the above pipeline generates a target vocabulary through term frequencies, which does not take into account how often/rarely these terms appear in other keyword corpora. For this, we replicate the above pipeline with two subtle but important differences to encourage more the niche target terms: *(i)* we obtain a term-document matrix capturing raw TF-IDF counts instead of TF counts, and *(ii)* we exclude the words that appear in more than 50 percent of all the keyword corpora (instead of 80 percent), further emphasizing niche domain-specific terminology.

**Examples**  Table 1 illustrates the target vocabularies produced by both variants for two keywords from the `CS.AI` domain. The TF vocabulary captures broadly relevant domain terms, while the TF-IDF vocabulary surfaces rarer, more specialized terminology. The full table with additional keywords is provided in Supplementary Table 5.

## 2.5 Prompt-Target Pair Construction

Our prompt-target pair construction pipeline utilizes keywords (Section 2.2), corresponding TF and TF-IDF vocabulary (Section 2.4), and the preprocessed sentences to extract prompts and targets for a domain of interest. To create prompts that effectively test model knowledge of each keyword, we first retrieve the preprocessed sentences that have a cosine similarity of at least 0.5 with the keyword. Further, to ensure that prompts provide enough context to predict the corresponding target, we filter out sentences shorter than 10 tokens (for the GPT-2 XL tokenizer) and 40 characters. Sentences containing no words from the target vocabulary after the 10th token are also excluded. Note that due to abundant data, these filtering steps still yield a large number of samples per keyword.

Following this procedure, the prompts consist of all tokens in a sentence up to a term from the target vocabulary, and the targets are the domain-specific terms following the prompts. This process is repeated for each keyword in the corpus until the desired number of prompts is reached or no further sentences remain for that keyword. Prompts and targets are constructed separately for the TF and TF-IDF vocabularies and are evaluated independently. For each keyword, 50 prompts and targets were created for both TF and TF-IDF cases to represent the diversity of the domain. Table 2 shows representative examples from two domains; a complete set is provided in Supplementary Table 8.

## 2.6 Model Evaluation through Prediction Ranks and Probabilities

Utilizing prompt-target pairs, we evaluate each model's ability to predict the target given only the preceding context. For this, input the model with the prompt and record the *probability* and the *rank* of the correct first token. If the target consists of multiple tokens, we employ a sequential evaluation approach: we first evaluate the rank of the first target token given the original prompt. For subsequent target tokens, we augment the prompt with the actual target tokens (not the model's predictions) to provide proper context. This ensures we measure the model's ability to complete the intended target sequence rather than penalizing it for alternative, potentially valid first tokens. We then average ranks and probabilities across all target tokens. Our corpus-based construction ensures that targets represent frequently occurring domain-specific terms; when multiple valid completions exist, models with sufficient domain knowledge assign high probability to multiple appropriate terms, keeping average ranks stable.

Table 2: Representative prompt–target pairs for selected keywords from two domains.

| Keyword | Prompt | Target |
|---|---|---|
| **Computer Science – Artificial Intelligence (`CS.AI`)** | | |
| Machine Learning | One approach, which has recently been immensely successful in machine learning with large artificial neural networks, is the idea of learning through gradient descent using the | backpropagation |
| Reinforcement Learning | Prior attempts at improving data efficiency in reinforcement learning, involved the use of an Experience | Replay |
| Deep Learning | From a different perspective, Learning Important Features Through Propagating Activation Differences introduced | DeepLIFT |
| **Quantitative Biology – Populations and Evolution (`Q-Bio.PE`)** | | |
| Phylogenetic Tree | In 2004, Speyer and Sturmfels showed a space of phylogenetic trees with a given set of labels on their leaves is a | tropical |
| Disease Spread | This finding is consistent with the theory of infectious disease spread in highly coupled | metapopulations |
| Evolutionary Dynamics | In the context of evolutionary dynamics, much of the quantitative work has been strongly inspired by two famous analogies: Fisher's tentative link between natural selection and the second law of | thermodynamics |

**Rank as our primary metric** We employed both rank and probabilities for model comparison, and empirically observed that probabilities are not well-calibrated. This observation aligns with the findings that show raw token probabilities are poorly calibrated in LLMs (Jiang et al., 2021), and this miscalibration is further exacerbated by instruction tuning and RLHF (Kadavath et al., 2022), making probability-based comparisons between base and chat models unreliable. In contrast, rank depends only on the relative ordering of the model's predictions and is invariant to the sharpness of the output distribution, isolating what the model knows from how confidently it expresses it. Hence, we adopt prediction rank as our primary metric.

**Statistical methodology** Predicted ranks can range from one to the full vocabulary size of the model, and we empirically observe that they may contain outliers across all evaluated models. To balance robustness to outliers with statistical efficiency (Rosenberger & Gasko, 1983; Wilcox & Keselman, 2003), we report the *20% trimmed mean* of predicted ranks, discarding the lowest and highest 20% of values. More specifically, we aggregate ranks and probabilities over all prompts and keywords within each domain, and report corresponding 95% confidence intervals. We assess relationships between metrics using Pearson correlation coefficients and their associated p-values to quantify statistical significance. For completeness, we also include untrimmed mean ranks and probabilities in the Appendix; while these exhibit similar trends, their absolute values substantially differ due to outliers.

## 3 Evaluation Setup

To evaluate the efficacy of our pipeline, we compare the *prediction rank*s produced by our benchmark against a reference, standard benchmark. To choose the reference benchmark, we experimented with widely used multiple-choice question benchmarks. However, we replicated their reported limitations, detailed in Sec. 3.1. To address these shortcomings, we establish a three-tier validation hierarchy consisting of a human-curated expert benchmark, a Claude-generated benchmark, and our automated pipeline (Sec 3.2).

### 3.1 Multiple-Choice Question Benchmarks Yield Inconsistent Evaluations

We show the inherent biases of benchmarking based on MCQ on the popular MMLU dataset. We test various prompting strategies, including shuffling answer choices and fixing the correct answer to a specific letter position (A, B, C, or D). As shown in Fig. 3 (left), these changes consistently alter the ranking of models (please see Sec. A.7.2-A.7.3 for detailed analysis). This sensitivity highlights the instability of chat

model evaluations through MCQs and reveals that the findings may be confounded by prompt engineering instead of solely measuring knowledge differences (Bai et al., 2022; Gupta et al., 2024; Alzahrani et al., 2024). Hence, MCQ benchmarks should not serve as the reference to validate our pipeline. Instead, we opt for completion-based benchmarks as our reference standard since they align with the training objective of both base and chat models, which we elaborate on in the next subsection.

### 3.2 A Three-Stage Validation of Our Pipeline

**Validation hierarchy**  We validate our pipeline through a three-stage hierarchy: a human-curated expert benchmark that serves as the ground truth, Claude-generated benchmarks as a scalable proxy for the expert benchmark, and finally, our automated pipeline. Our validation strategy proceeds in two steps. First, we measure how well the ranks computed on a Claude-generated benchmark correlate with those computed on a manually curated reference standard. The high correlation implies that Claude-generated benchmarks are reliable proxies for expert human judgment. Second, we show on all four domains that our automated pipeline correlates strongly with the Claude benchmark. Together, these two results imply that our pipeline produces evaluations consistent with expert-level domain knowledge assessment.

**Expert benchmark**  We created a comprehensive manual expert benchmark from the 2025 edition of the textbook "Understanding Deep Learning" (Prince, 2023), extracting 281 prompt-target pairs from this authoritative source. This benchmark serves as our ground truth, as it reflects domain expertise independent of any LLM. As manual curation requires substantial domain expertise and time, we restrict this benchmark to our primary development domain (CS.AI), where we can most reliably assess its quality. Examples from the expert benchmark are shown in Table 7.

**Claude benchmark**  We provide the SOTA Claude Sonnet 4 model (Anthropic, 2025) with the top 20 keywords generated by our pipeline (Section 2.2) and request 50 prompts per keyword with corresponding targets (template in Appendix 14). Table 3 presents example prompts for three CS.AI keywords.

**Results**  The right panel of Fig. 3 demonstrates that the Claude-generated benchmark exhibits strong correlation with the manually curated expert benchmark (r=0.91, p=0.012), validating that Claude can serve as a scalable proxy for expert human curation. Crucially, our TF-based pipeline shows even stronger alignment with the expert benchmark (r=0.99, p<0.001), establishing that our automated approach captures domain expertise patterns consistent with expert judgment. Because manual validation is only feasible for our primary development domain (CS.AI), we use the Claude benchmark as the reference standard in all subsequent experiments (Section 4). The strong pipeline-to-Claude correlations we observe across all domains (explained in detail in the next section), combined with this validated pipeline-to-expert alignment in CS.AI, support the conclusion that our pipeline produces reliable evaluations across diverse domains.

## 4 Results and Discussions

To validate the efficacy and robustness of our proposed pipeline, we conduct a comprehensive set of experiments across diverse domains. First, in a controlled domain adaptation setup, we show the robustness of our *prediction rank* metric (Sec.4.1). We then evaluate model checkpoints during general pretraining (Sec.4.2) and continual pretraining (Sec.4.3), revealing that our benchmark captures nuanced patterns of knowledge acquisition that are missed by perplexity and attribution rate. Finally, we apply our benchmark to compare base and instruction-tuned models (Sec. 4.4), showing its consistency across heterogeneous model families and four diverse domains without relying on few-shot protocols. We remind that we use the same parameters in our pipeline in all experiments. The results stay consistent across all experiments, indicating the robustness and generalizability of our approach to various input corpora.

**Baseline metrics**  Throughout our experiments, we compare our prediction rank metric against two possible alternatives: perplexity and last layer attribute rate. For perplexity calculation, we employed the Semantic Scholar portion of M2D2 (Reid et al., 2022), a multi-domain corpus of scientific papers already cleaned for domain adaptive language model pretraining. More specifically, we measure a model's knowledge on a domain by computing its perplexity on the corresponding M2D2 portion. We additionally explore the *attribute rate*

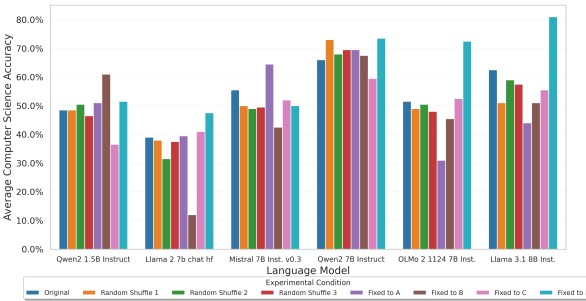
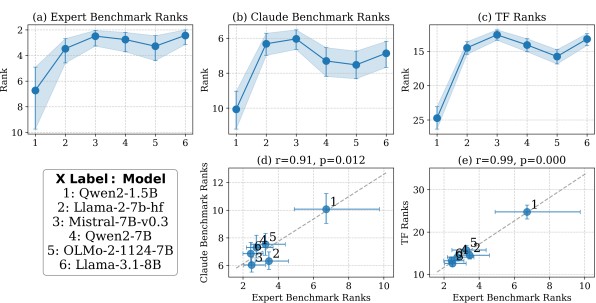

Figure 3: Validation of benchmarking approaches. Left: MCQ benchmarks show significant sensitivity to option ordering. Right: Completion-based validation across six base models showing the ranks computed on (a) manual expert benchmark, (b) Claude-generated benchmark, and (c) TF-based pipeline. Correlation analysis reveals (d) r=0.91, p=0.012 between Claude and expert benchmarks, and (e) r=0.99, p<0.001 between TF-based and expert benchmarks, validating that Claude-generated benchmarks reliably capture expert-level domain knowledge patterns.

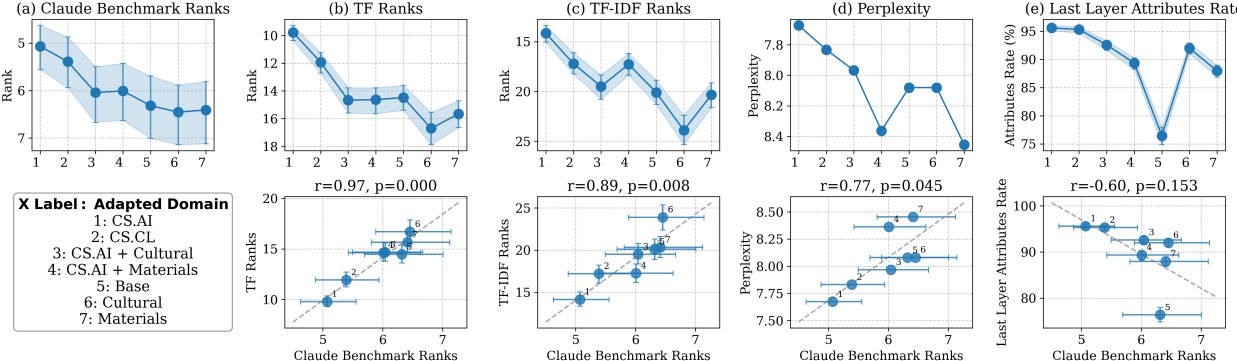

Figure 4: Validation of our pipeline through domain adaptation, where we adapt Llama-2-7B to seven domains separately. *Top row:* The x-axes show the adapted domains ordered by proximity to `CS.AI` and the y-axes represent evaluation metrics. Prediction ranks on (a) the reference benchmark generated by Claude Sonnet 4, and (b-c) TF- and TF-IDF-based benchmarks produced by our pipeline, along with baseline metrics (d) perplexity and (e) last-layer attribution rate. *Bottom row:* Correlation analysis between each metric and the Claude benchmark. Claude, TF and TF-IDF ranks follow the expected pattern, where models adapted to domains similar to `CS.AI` achieve better rank than those trained on unrelated domains. We observe significantly strong correlations between ranks on Claude-generated reference dataset and our TF-based methods (r=0.97, r=0.89) while perplexity and attribution rate show weaker correlations.

metric recently proposed to study factual knowledge in LLMs (Geva et al., 2023). The attribute rate for a prompt quantifies the overlap between model-generated completions given a prompt and tokens related to the subject of the prompt (see Sec. A.3 for a detailed explanation). Layer-wise attribute rate has provided deep insights into knowledge extraction through the layers. We employ the attribute rate of the last layer to quantify the entire knowledge within a network.

## 4.1 Validation of Our Pipeline through Domain Adaptation

To validate our pipeline, we conduct controlled domain adaptation experiments where we adapt Llama2-7B models to domains with varying semantic proximity (`CS.AI`, `CS.CL`, `Cultural`, `Materials`, `CS.AI + Cultural`, `CS.AI + Materials`) to our target evaluation domain (`CS.AI`). As shown in Fig. 4, models adapted to semantically similar domains (e.g., `CS.CL`) consistently achieve better prediction ranks on our TF- and TF-IDF-based benchmarks compared to models trained on unrelated domains (e.g., `Material Science`), demonstrating that our evaluation captures genuine domain expertise. Our benchmarks show

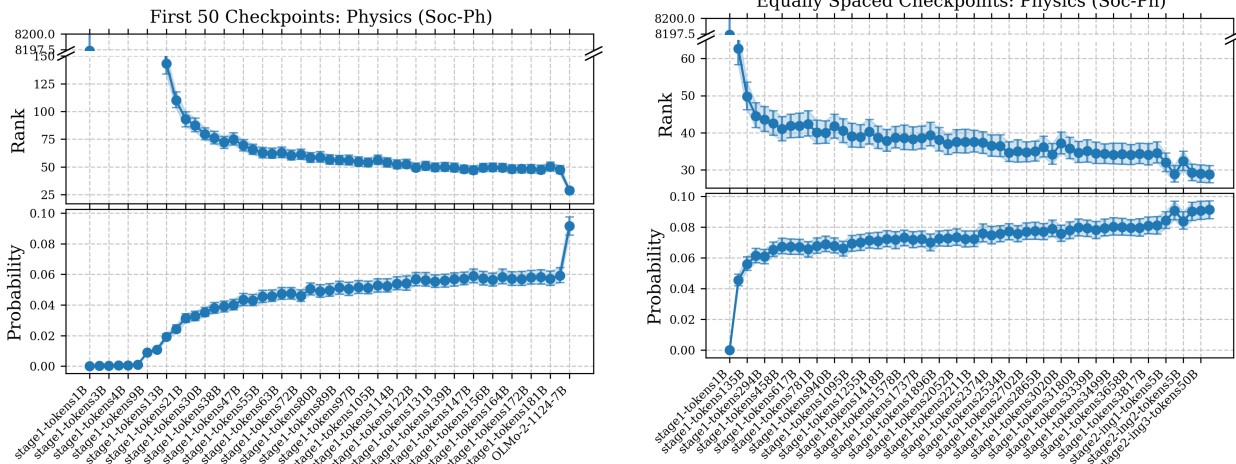

Figure 5: Prediction ranks and probabilities of OLMo-2 pretraining checkpoints on the `Physics and Society domain (Physics (Soc-Ph))`. The left figure displays the results across the first 50 checkpoints, with the last point representing the final model for reference. The right figure displays equally spaced checkpoints from the entire pretraining. Consistent patterns above demonstrate that our benchmark can guide the knowledge accumulation for a domain of interest by replacing the average perplexity metric.

strong correlation with the reference benchmark, while perplexity shows weaker correlation and attribution rate shows no correlation. These controlled experiments validate that our pipeline effectively quantifies domain-specific knowledge acquisition.

**Supplementary findings** We repeat the same experiment with GPT2-XL and observe the same results. Complete experimental setup, detailed results for GPT2-XL, correlation analyses, and all evaluation metrics are provided in Sec. A.5 and Fig. 8. Notably, the larger Llama2-7B model achieves much better absolute ranks compared to GPT2-XL, with relatively smaller changes across domain adaptative training runs since bigger models already possess substantial knowledge and exhibit less dramatic adaptation effects. Additional analyses including semantic similarity validation and mean aggregation results are in Sec. A.6 and Sec. A.9.

## 4.2 Quantifying Knowledge Acquisition During General Pretraining

Recent work employed MCQs to monitor performance during pretraining (OLMo et al., 2024). Notably, evaluation logs[1] reveal that downstream performance on 16 MCQ tasks saturates halfway through training, even as perplexity continues to improve. This suggests that downstream MCQ evaluation can behave inconsistently, or even misleadingly, when used as a proxy for overall model learning.

As an alternative, we propose to quantify knowledge acquisition by evaluating the intermediate checkpoints on our TF-based benchmark. Fig. 5 presents the rank and probabilities obtained on the `Physics and Society domain (Physics (Soc-Ph))` domain. On the left panel, we visualize the first 50 checkpoints (roughly 5% of the entire pertaining). We observe a significant improvement in model ranks during initial pretraining, which is expected as the weights evolve from their randomly initialized state and the model sharply improves. Notably, even at the 50th checkpoint, a significant difference in rank persists between this intermediate state and the final base model, indicating substantial potential for further training in this domain.

For the second analysis, we selected two groups of checkpoints *(i)* 50 equally spaced checkpoints from the first stage of pretraining on 4T tokens, and *(ii)* 5 checkpoints from the second stage on 150B high-quality data (Supplementary Table 10 lists all evaluated checkpoints.) The right panel of Fig. 5 illustrates an improvement trend throughout pretraining, which much more gradually saturates than MCQ evaluations. Finally, we observe a more clear improvement when switched to high-quality data (the last 5 checkpoints).

---

[1] https://wandb.ai/ai2-llm/OLMo-2-1124-7B/reports/OLMo-2-7B-Nov-2024--VmlldzoxMDUzMzE1OA

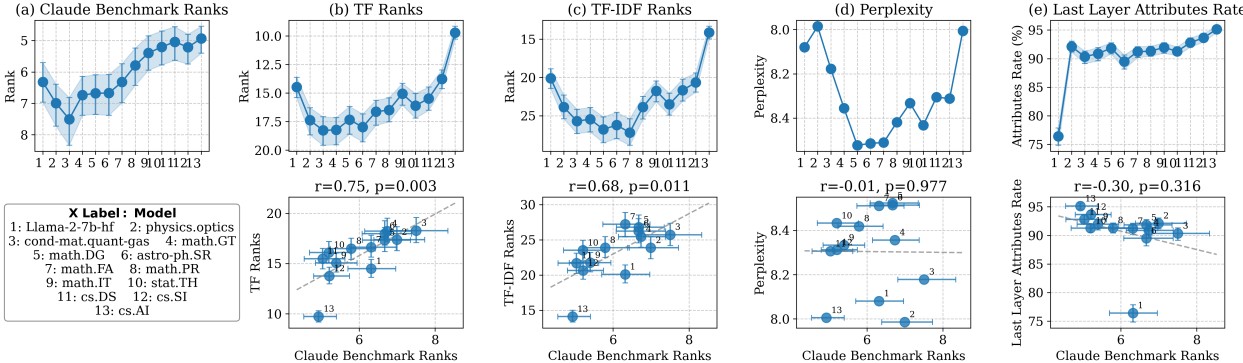

Figure 6: Evaluation results for 12 checkpoints from continual pretraining of Llama2-7B, along with the base model. The bottom-left panel displays the sequence of pretraining domains in the order they were introduced during training. *Top row:* Prediction ranks on *(a)* the reference benchmark generated by Claude Sonnet 4, and *(b–c)* TF- and TF-IDF-based benchmarks produced by our pipeline. The last two columns show the baseline metrics *(d)* perplexity and *(e)* last-layer attribution rate. *Bottom row:* Correlation of each metric with the Claude benchmark. While ranks computed on our benchmarks align with the similarity between pretraining domains and the target CS.AI domain, perplexity and attribute rate do not offer robust measures for knowledge acquisition.

**Supplementary findings** We also evaluated the checkpoints on the CS.AI domain. The similar trends depicted in Supp. Fig. 16 verify the consistency of our evaluation across downstream domains. We note that the overall ranks are considerably better on CS.AI, indicating the relative ease of the task for the model. We attribute this to relatively abundant training data from this domain, resulting in improved performance.

### 4.3 Continually Pretraining Llama2-7B model

To obtain an expert model for a specific domain, continual pretraining on related domains has been shown to outperform adaptation solely on the target domain (Gururangan et al., 2020). However, recent work (Yıldız et al., 2024) highlights that poor selection of pretraining corpora can degrade performance. This underscores the importance of tracking knowledge acquisition throughout continual learning. In this section, we show that perplexity provides an uncalibrated and often misleading proxy for model knowledge, whereas our pipeline enables more robust and interpretable analyses.

**Training setup** Following previous experiments, we select CS.AI as the target domain and continually pretrain Llama2-7B on a sequence of domains from M2D2. To construct a meaningful training order, we randomly sample 5000 abstracts from arXiv, embed them using Sentence Transformers, and compute cosine similarity between their embeddings and those of CS.AI domain abstracts. We then sort the M2D2 domains by their similarity to CS.AI. From this list, we uniformly sample 12 domains, sort them by *dissimilarity* to CS.AI, and use them as the continual pretraining sequence for Llama2-7B (see domain names in Fig. 6).

**Findings** Fig. 6 shows how different metrics evolve as the model is continually pretrained. The ranks on the Claude benchmark (Fig. 6a) initially decrease, then remain relatively stable, and eventually increase. This progression aligns with the increasing similarity between the continual pretraining domains and the target CS.AI domain. The prediction ranks on our TF- and TF-IDF-based benchmarks follow a similar trajectory (Figs. 6b and 6c), whereas perplexity and attribute rate show no clear correlation with Claude benchmark ranks (Figs. 6d and 6e). Taken together, our framework offers a more robust and informative assessment of changes in performance during continual pretraining than the commonly used scores.

**Supplementary findings** For completeness, we report target token probabilities (Fig. 17) and untrimmed mean aggregation (Figs. 18- 19). Across all evaluations, the patterns discussed above remain consistent.

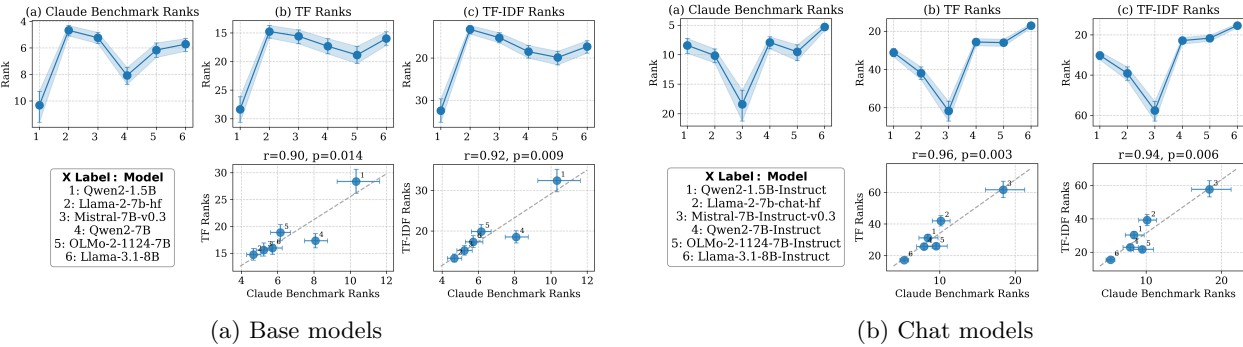

(a) Base models                                  (b) Chat models

Figure 7: Evaluation of 6 base models (left) and 6 chat models (right) on `General Economics (Econ-GN)` domain. The top row shows prediction ranks, and the bottom row shows the correlations between the Claude benchmark and our TF- and TF-IDF-based benchmarks.

## 4.4 Comparing Base and Chat Models

Up to this point, our analysis has focused on targeted settings such as adapted models, general-purpose model pretraining, and continual pretraining. We now extend our evaluation to six publicly available open-weight models spanning a range of scales and architectures. For each model, we include both the base and chat variants. This setup enables a unified evaluation framework that facilitates direct comparisons across model families, without relying on conventional MCQ benchmarks; thus avoiding the issues discussed in Sec. 3.1.

Fig. 7 compares six base models and their aligned (instruction-tuned) counterparts on the `General Economics (Econ-GN)` domain. Most notably, we observe a strong correlation between the ranks computed on the Claude benchmark and our TF- and TF-IDF-based benchmarks. While perplexity also correlates well with Claude ranks for base models (Fig.20), it fails to provide reliable signals for aligned models. In contrast, our rank-based approach remains consistent across both base and aligned checkpoints, demonstrating its superiority as a general-purpose evaluation metric. Please see Figs 21,22,23 for the same analysis repeated on three other domains, leading to consistent results.

Our rank analysis also allows us to inspect how alignment impacts model knowledge. Fig. 24 reveals that base models generally outperform their chat counterparts, corroborating findings from Bai et al. (2022) who demonstrated that smaller models experience significant "alignment tax" where performance degrades after instruction tuning. This degradation is particularly pronounced for Llama2-7B and Mistral-7B-v0.3, whose performance drop after instruction tuning across all domains, suggesting significant room for improvement in their alignment pipelines. Moreover, our framework provides a more fine-grained view than the generic evaluations in Bai et al. (2022), revealing that some models like Qwen2-1.5B and Llama3.1-8B exhibit smaller degradations or even occasional improvements, demonstrating that alignment effects are highly heterogeneous across architectures and domains.

## 5 Related Work

The evaluation of LLMs evolved from simple statistical measures to sophisticated benchmarking frameworks, yet challenges remain for domain-specific assessment. Initially, perplexity served as the primary evaluation metric (Jelinek et al., 1977; Gururangan et al., 2021; Yıldız et al., 2024). Recent work revealed that perplexity primarily captures general linguistic capabilities rather than domain knowledge (Xu et al., 2024; Meister & Cotterell, 2021; Öncel et al., 2024). Alternative approaches like attribute rate (Geva et al., 2023) show inconsistent patterns across model architectures.

Alternative line of research introduces MCQs benchmarks, e.g., MMLU (Hendrycks et al., 2020), GLUE (Wang et al., 2018), and BIG-bench (Srivastava et al., 2022). A series of works revealed their pitfalls that few-shot performance conflates model size with domain knowledge (Brown et al., 2020), benchmarks show sensitivity to formatting and option ordering (Alzahrani et al., 2024; Gupta et al., 2024), alignment processes introduce additional biases (Bai et al., 2022), and MCQ evaluations saturate during pretraining

while perplexity continues improving (OLMo et al., 2024). Critically, existing benchmarks focus on broad, mixed-domain tasks rather than specific domains, creating challenges for selecting base models for domain adaptation (Han et al., 2023; Cui et al., 2025; Nguyen et al., 2023). The prevalence of MCQ formats over completion tasks (Chang et al., 2024) further misaligns with base model training objectives.

Recent efforts to automate benchmark generation follow two approaches. LLM-based generation (Perez et al., 2023; Yuan et al., 2025) addresses curation costs but faces limitations in long-context processing (Li et al., 2024), grounding without parametric bias (Monea et al., 2023), and computational efficiency (Chavan et al., 2024). Curated automated systems like LiveBench (White et al., 2024) and LiveCodeBench (Jain et al., 2024) mitigate contamination through recent sources or existing problem collections, but require structured data sources (news articles, programming contests) and cannot generate benchmarks from arbitrary raw domain corpora. Existing domain-specific benchmarks like MedQA (Jin et al., 2021) and LegalQA (Louis et al., 2024) organize around broad disciplinary categories rather than research subdomains, employ evaluation formats (MCQs, long-form answers) incompatible with base models, and lack mechanisms for subdomain-specific assessment or contamination-resistant updates.

Our proposed pipeline converts raw domain corpora into completion-style benchmarks, addressing these limitations through: *(i)* unified evaluation for base and chat models, *(ii)* domain-specific rather than general linguistic assessment, *(iii)* elimination of prompt sensitivity biases, and *(iv)* scalable evaluation across arbitrary domains, and *(v)* contamination-resistant evaluation through on-demand generation from recent corpora.

# 6 Conclusion

We introduced and validated a framework to construct completion-style benchmarks from raw domain corpora. We generate prompt–target pairs from automatically extracted domain keywords and related vocabularies, and quantify model expertise through prediction ranks of the targets given the prompts. Strong correlation with manually curated expert benchmarks (r=0.91-0.99, p≤0.012) confirms our automated approach captures domain expertise patterns consistent with human expert judgment. Our lightweight evaluation framework circumvents the intrinsic biases of multiple-choice questions and does not suffer from benchmark contamination. Likewise, it allows for automatically updating a benchmark with ever-growing domain text with minimal cost.

Our evaluations demonstrate that this framework captures meaningful distinctions across models and training settings: it reliably traces knowledge acquisition during general, adaptive, and continual pretraining and highlights performance differences between base and aligned models. Altogether, our work offers a principled alternative for practitioners and researchers seeking to assess and track LLM competence in specific domains, while avoiding benchmark contamination and being applicable to both base and instruction-tuned models.

# 7 Limitations

While our proposed pipeline demonstrates strong empirical results across multiple experimental settings, several limitations require discussion. Our validation approach uses benchmarks generated by Claude Sonnet 4 as a reference standard. Although we validate this choice with a manual expert benchmark for `CS.AI` (r=0.91, p=0.012), extending this manual validation to all four domains was beyond the scope of this work. The strong pipeline-to-Claude correlations across all domains, combined with intuitive patterns in controlled experiments, suggest that this validation generalizes beyond `CS.AI`.

Our experimental scope is limited to arXiv academic texts across four STEM domains, and computational constraints restricted model coverage to eight base and six chat variants, ranging from 1.5B to 8B parameters. While evaluation of larger models (e.g., >70B) would provide additional validation, consistent patterns across diverse architectures (GPT2, Llama2, Mistral, Qwen2) support broader applicability.

Finally, chat models are optimized for conversational interaction rather than completion tasks, which could affect our evaluation. However, completion tasks align with the pretraining objective shared by both model types, and the systematic patterns we observe suggest our method captures meaningful signals about knowledge retention across both settings.

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

# A   Appendix

Table 3: Prompts and targets generated by Claude Sonnet 4 for three keywords from `CS.AI`

Prompts and Targets for Keywords from `CS.AI`

**Reinforcement Learning Agents:**
**Prompt:** The entity that takes actions in an environment to maximize reward is known as an
**Target:** agent
**Prompt:** The decision-making component in reinforcement learning systems is termed an
**Target:** actor

**Policy Optimization:**
**Prompt:** The process of improving action-selection strategies through gradient ascent is known as
**Target:** gradient optimization
**Prompt:** The technique of updating policies using importance sampling is termed
**Target:** importance sampling

**Explainable AI:**
**Prompt:** Methods that make AI decision-making processes interpretable are called
**Target:** explainable methods
**Prompt:** The process of revealing the reasoning behind AI outputs is called
**Target:** reasoning explanation

## A.1   Details of Our Keyword Generation Pipeline

Here, we detail our keyword generation pipeline.

**Initial preprocessing**   First, we standardize the abstracts by converting them to lowercase, removing bracketed content, handling apostrophes and contractions, and implementing custom tokenization to ensure consistency. Our custom tokenization preserves hyphenated words (e.g., fine-tuned) to maintain the integrity of technical terms.

**Building and validating n-grams**   Next, we build n-grams from size 2 to 7 using Gensim (version 4.3.3) Phrases model (Rehurek & Sojka, 2010; Mikolov et al., 2013). For simplicity, we have employed default parameters for a minimum count threshold of 5 occurrences and a collocation threshold of 10 to identify meaningful word combinations. We further ensure the quality and relevance of keywords by including words that contain only alphabetic characters and hyphens (e.g., "deep-learning" is valid, "model2" is not).

**Length-based filtering of the keywords**   The third stage implements adaptive thresholding to select only the most informative keywords. It allows us to determine the proportion of each n-gram to keep in our keyword list. For our work, we have selected proportions of 50% bigrams, 30% trigrams, 15% four-grams, and 5% five-or-more-grams based on empirical analyses. If a domain had n-grams below the given proportion, then the ratios are adaptively adjusted based on the actual proportion of each n-gram length in the corpus. For our corpus, we targeted approximately 300 keywords from each domain.

**Redundancy elimination**   The final stage is the semantic redundancy reduction, for which we compute the cosine similarity between all keyword pairs using the "all-MiniLM-L6-v2" (Wang et al., 2020) model from

Table 4: Ten keywords generated by our pipeline for each of the four domains used in this paper.

| DOMAIN | KEYWORDS |
|---|---|
| Computer Science – Artificial Intelligence (CS.AI) | machine learning, reinforcement learning, deep learning, explainable ai, graph neural, transfer learning, generative adversarial, imitation learning, causal inference, continual learning |
| Physics – Physics and Society (Physics (Soc-Ph)) | complex network, epidemic dynamics, community detection, opinion dynamics, information diffusion, scale-free network, human mobility, temporal networks, evolutionary dynamics, network science |
| Quantitative Biology – Populations and Evolution (Q-Bio.PE) | evolutionary dynamics, disease spread, phylogenetic tree, population dynamics, natural selection, mutation rate, genetic diversity, infectious disease, fitness landscape, fixation probability |
| General Economics (Econ-GN) | economic growth, monetary policy, income inequality, labor market, nash equilibrium, climate change, financial market, international trade, fiscal policy, unemployment rate |

Sentence Transformer library (Reimers & Gurevych, 2019). We identify pairs whose similarity score passes 0.85 as *duplicates* and merge them into a single keyword. We maintain the shorter string name (e.g., "complex network" over "complex networks"). An example keyword list from the computer science - artificial intelligence (CS.AI) domain is presented in Supplementary Table 5.

## A.2 Matching Keywords and Sentences: Implementation Details

We first strip the entries in RedPajama of excess whitespace and then convert these documents into individual sentences using spaCy's "en_core_web_sm" model (Honnibal et al., 2020). Subsequently, we embed all sentences as well as keywords obtained in Section 2.2 and calculate the cosine similarity between the sentence and keyword embeddings. For each keyword, we set a similarity threshold of 0.5 between the keyword and the corresponding sentence to extract only the sentences semantically related to that keyword.

Selected sentences further undergo a systematic cleaning process. First, any LaTeX-specific formatting was converted to plain text using LatexNodes2Text (Faist, 2021). We removed all citations using expression patterns targeting common citation commands (e.g., \citet) and converted tabs, newlines, and other whitespace to single spaces. We further validated characters against a predefined set of allowed characters (ASCII letters, digits, whitespace, and punctuation) to remove special characters. We removed any sentence that failed the character validation from further analyses. In the end, we obtain a clean mapping of sentences for each keyword in a domain of interest.

## A.3 Attribute Rate Calculation

We adopt the attribute rate analysis introduced by Geva et al. (2023) to understand knowledge progression through layers in LLMs. This metric quantifies the subject-related information contained in each layer's representations by measuring the overlap between model-generated tokens and tokens semantically related to the input subject. In our work, subjects refer to the keywords we extracted in Section 2.2.

### A.3.1 Token Analysis Framework

Before calculating attribute rates, we must first establish the set of tokens semantically related to each keyword. For a given keyword, we aggregate all clean sentences obtained in Section 2.3 to create a unified keyword corpus. We then tokenize these sentences using the corresponding model tokenizer and apply the following filtering process:

1. Remove all tokens associated with stopwords (using NLTK stopwords list)

2. Remove tokens with character length of less than three

3. Maintain a frequency counter for the remaining filtered tokens

4. Preserve the top 1200 most frequent tokens for each keyword (to be consistent with the average tokens employed in Geva et al. (2023))

This process yields a set $A_s$ containing the most relevant tokens for keyword $s$, which serves as our ground truth for semantic relatedness in the attribute rate calculation.

### A.3.2 Layerwise Attribute Rate Methodology

For intermediate layers, the attribute rate computation involves projecting hidden representations to vocabulary space and measuring overlap with our established token sets. Following Geva et al. (2023), given a keyword $s$, we calculate the attribute rate at layer $\ell$ and position $i$ as follows:

First, we project the hidden state to vocabulary space:

$$\text{projs} = h_i^\ell \cdot E^T \tag{1}$$

where $E$ is the model's input embedding matrix and $h_i^\ell$ is the hidden state at position $i$ and layer $\ell$.

Next, we extract the top-$k$ tokens by removing stopwords and tokens with fewer than 3 characters from the projected vocabulary distribution. Finally, we calculate:

$$\text{Attribute Rate} = \frac{|\text{Top-}k \text{ tokens} \cap A_s|}{k} \times 100\% \tag{2}$$

We focus primarily on the last token position of each keyword, as it can attend to all preceding subject tokens and thus contains the most comprehensive subject representation (Geva et al., 2023). Note that in the original implementation, for the final layer, Geva et al. (2023) used the output of the final layer normalization, which corresponds essentially to the model's actual output logits.

### A.3.3 Cross-Architectural Last Layer Attribute Rate

While layerwise attribute rate provides deep insights into knowledge progression, it has primarily been employed for GPT family models (nostalgebraist, 2020; Dar et al., 2022; Geva et al., 2023) and faces challenges with newer architectures like Llama-2, Llama-3.1, and OLMo-2 due to architectural differences that make direct projection of hidden states potentially incomparable across models.

Since the original method shows that attribute rate follows a monotonic increasing pattern through the layers (Geva et al., 2023), with the final layer representing the culmination of the subject's enrichment throughout the model, we can focus exclusively on the final layer output for cross-architectural comparison. For the last layer attribute rate, we directly use the model's output token probabilities:

$$\text{LLAR} = \frac{|\text{Top-}k \text{ clean output tokens} \cap A_s|}{k} \times 100\% \tag{3}$$

where LLAR is an abbreviation for last layer attribute rate, and Top-$k$ clean output tokens are the top-$k$ tokens from the model's final output distribution after removing stopwords and tokens with fewer than 3 characters. We use $k = 50$ in our analysis to maintain consistency with prior work while ensuring sufficient coverage of the model's predictions.

This approach leverages the fact that the final layer projection in the original methodology essentially captures the same information as the model's direct output, making it robust across different architectures while maintaining the theoretical foundation established by Geva et al. (2023). By focusing on the last layer, we obtain a metric that enables meaningful comparisons across different model architectures while capturing the final state of factual knowledge extraction.

### A.4 Sample Document from Dataset

Below we present a sample document from our arXiv and Kaggle dataset after mapping, showing both the metadata and the beginning of the full-text content:

---

**Sample Document: arXiv ID 1607.04768**

**Title:** Hoffmann-Ostenhof's conjecture for traceable cubic graphs
**Categories:** math.CO cs.DM
**Abstract:** In 2011 Hoffmann-Ostenhof proposed the following conjecture: Every connected cubic graph can be decomposed into a spanning tree, a matching and a family of cycles. In this paper we prove that this conjecture holds for traceable cubic graphs.

**Introduction:** Let $G$ be a simple undirected graph with the *vertex set $V(G)$* and the *edge set $E(G)$*. A vertex with degree one is called a *pendant vertex*. The distance between the vertices $u$ and $v$ in graph $G$ is denoted by $d_G(u, v)$. A cycle $C$ is called *chordless* if $C$ has no *cycle chord* (that is an edge not in the edge set of $C$ whose endpoints lie on the vertices of $C$). The *Induced subgraph* on vertex set $S$ is denoted by $\langle S \rangle$. A path that starts in $v$ and ends in $u$ is denoted by $\widehat{vu}$. A *traceable* graph is a graph that possesses a Hamiltonian path. In a graph $G$, we say that a cycle $C$ is *formed by the path $Q$* if $|E(C) \setminus E(Q)| = 1$. So every vertex of $C$ belongs to $V(Q)$...

---

Our combined dataset contains 1.56 million documents with similar structures, covering the full spectrum of scientific disciplines represented in arXiv.

### A.5 Validation of Our Pipeline by Adapting GPT2-XL to Diverse Domains

In this section, we conduct controlled domain adaptation experiments, where model training domains vary in semantic proximity to the evaluation domain (CS.AI). We expect models adapted to domains closely related to AI to outperform those trained on unrelated domains, thereby demonstrating that our benchmark is sensitive to domain-relevant knowledge.

**Adapted domains**  We train the GPT2-XL model on six different domain-specific subsets of M2D2: *(i)* two subdomains of computer science (Artificial Intelligence (CS.AI), and Computation and Language (CS.CL)); *(ii)* two domains unrelated to computer science (Culture and Humanities, and Material Science); and *(iii)* a mix of these domains (CS.AI + Material Science and CS.AI + Culture and Humanities). These six adapted variants, along with the original base GPT2-XL model, form the basis of our evaluation.

**Evaluation sets**  We compute the prediction rank of these models on the Claude-generated benchmark on AI. Likewise, we construct prompt-target pairs by running our pipeline on CS.AI domain of the RedPajama-Data-1T dataset, where prompts and targets are separately obtained by TF and TF-IDF. We further computed the models' last layer attribution rate using the tokens obtained from the same dataset and the perplexity computed on the test set of CS.AI domain in M2D2. Since all evaluation sets are restricted to the AI domain, we can intuitively order expected model performances.

**Findings**  First, Fig. 8a confirms that models trained on domains similar to AI rank higher on the Claude benchmark, while training on unrelated domains degrades the rank. Ranks obtained on our TF- and TF-IDF-based benchmarks follow a very similar trend, evident by their strong correlation with the Claude benchmark (Fig. 8b-8c, bottom row). Perplexity (Fig. 8d) correlates less with the ranks obtained on the Claude benchmark compared to our pipeline. Finally, the last layer attribute rate in Fig. 8e shows no correlation at all. Altogether, these findings validate the efficacy of the Claude benchmark as well as our pipeline.

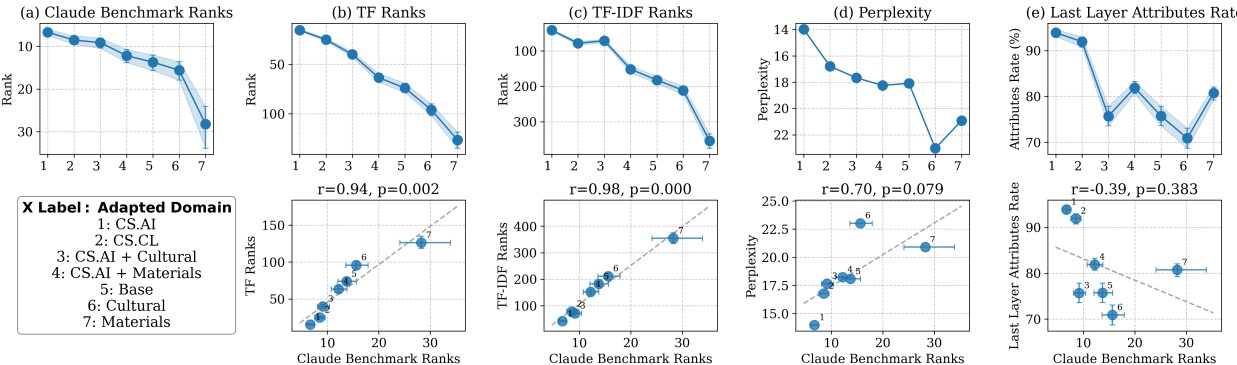

Figure 8: Evaluation of six domain-adapted GPT2-XL models and the base model. The bottom-left panel shows the adapted domains, ordered by their similarity to the target domain, CS.AI. Top row: Prediction ranks on *(a)* the Claude benchmark generated by Claude Sonnet 4, and *(b–c)* TF- and TF-IDF-based benchmarks produced by our pipeline. We also report baseline metrics: *(d)* perplexity and *(e)* last-layer attribution rate. Bottom row: Correlation of each metric with the Claude benchmark. TF- and TF-IDF-based ranks align closely with the Claude benchmark, whereas perplexity and attribution rate do not reflect expected trends in domain adaptation.

## A.6  Checkpoint Evolution During GPT2-XL Adaptive Pre-training

We compared our introduced benchmark and evaluation method for tracking the effects of domain adaptation for the GPT-2 XL model during training. We call the GPT-2 XL model the base model, whose metrics are considered as the baseline as shown by the black dotted horizontal line in Supplementary Figure 9, and the CS.AI final (red dotted line) represents the value of metrics corresponding to the final checkpoint on the model adapted to the CS.AI domain from the M2D2 dataset. Each model starts from either the base model or from a pre-adapted model and is trained on equal steps of data for fair comparison from the target domain of interest, because each domain has different amounts of data available in the M2D2 dataset, and each step corresponds to a batch size of 16. The start and end domain is mentioned in the figure legend. For example, "CS.AI to Materials" means the starting model was an adapted model on the CS.AI domain and is further trained in steps on data from the Materials Science domain. The results presented for the test conducted for these metrics on the CS.AI dataset from arXiv, except for perplexity, which was performed on the test dataset from the M2D2 domain.

We can see a consistent pattern, as expected, in all of the metrics. The domain with CS.AI as the target domain performs better than other adapted models, showing the dominance of the target domain over the initial domain. CS.CL (Computer Science – Computation and Language), as expected because of its proximity to the CS.AI domain, turns out to be the second-best performer. Models whose target ends at Materials Science seem to perform the worst out of all because of irrelevance to the CS.AI domain. When the target is fixed, we can see a similar pattern for the initial domain as well; for example, out of the models that have Materials Science as the target domain, the model started from the base GPT2-XL model has far worse rank, perplexity, and attribute rates than the one started from the CS.AI domain.

We can also observe that for smaller models like GPT2-XL, the domain adaptation happens rapidly, even with a small amount of data, as can be seen by a substantial jump in each metric for 500 steps of adaptive training. This implies that for smaller models, we can achieve significant improvement without the need to spend a high computational budget for domain adaptive training.

TF and TF-IDF model ranks seem to perform very similarly to the Claude benchmark. Perplexity, although on average displaying a similar trend, shows somewhat peculiar behavior for "CS.AI to Materials" and "Base to CS.CL" cases. We expect that the model that has been trained on a related domain should perform better than one trained on an irrelevant domain, which seems to be the case with all other metrics, demonstrating the limitations of perplexity compared to our robust method of rank evaluation from our introduced pipeline.

Table 5: Example of 3 keywords, 10 elements from word-list (TF and TF-IDF), and 15 tokens from tokenizers (GPT2-XL, LLAMA-2 7B, OLMo-2 7B) for CS.AI domain.

| KEYWORD | TF TARGET VOCABULARY | TF-IDF TARGET VOCABULARY | TOKEN LIST |
|---|---|---|---|
| machine learning | Backpropagation, Support, Vector, SVM, Random, Machines, Supervised, Bayes, Regression, Forest | Logistic, Naive, AutoML, Boosting, SVMs, kNN, XGBoost, AdaBoost, Scikit, Perceptron | GPT2-XL: learning, machine, data, algorithms, models, methods, supervised, classification, training, features, regression, neural, prediction, performance, support
LLAMA-2 7B: learning, machine, data, algorithms, models, super, vised, methods, classification, training, features, regression, neural, prediction, performance
OLMo-2 7B: learning, machine, data, algorithms, models, methods, supervised, classification, training, features, regression, neural, prediction, classifier, vector |
| reinforcement learning | Replay, Policy, rewards, actor, imitation, MDP, transition, cumulative, critic, Optimization | imitation, critic, Actor, Inverse, MDPs, shaping, replay, bandit, Bellman, Proximal | GPT2-XL: learning, reinforcement, policy, reward, agent, state, function, environment, optimal, actions, value, exploration, deep, training, decision
LLAMA-2 7B: learning, rein, cement, policy, reward, agent, state, function, environment, optimal, actions, value, deep, training, decision
OLMo-2 7B: learning, reinforcement, policy, reward, agent, state, function, environment, optimal, actions, value, exploration, deep, training, decision |
| deep learning | Convolutional, LSTM, speech, ImageNet, Recurrent, DNN, regularization, Adam, hardware, optimizer | ResNet, PyTorch, AlexNet, TensorFlow, SGD, ReLU, VGG, CIFAR, Boltzmann, Hinton | GPT2-XL: deep, learning, neural, networks, data, training, performance, architectures, tasks, classification, image, layers, features, algorithms, large
LLAMA-2 7B: deep, learning, neural, networks, data, architect, ures, training, performance, tasks, classification, image, layers, features, convolution
OLMo-2 7B: deep, learning, neural, networks, data, training, performance, architectures, tasks, classification, image, layers, features, convolution, algorithms |

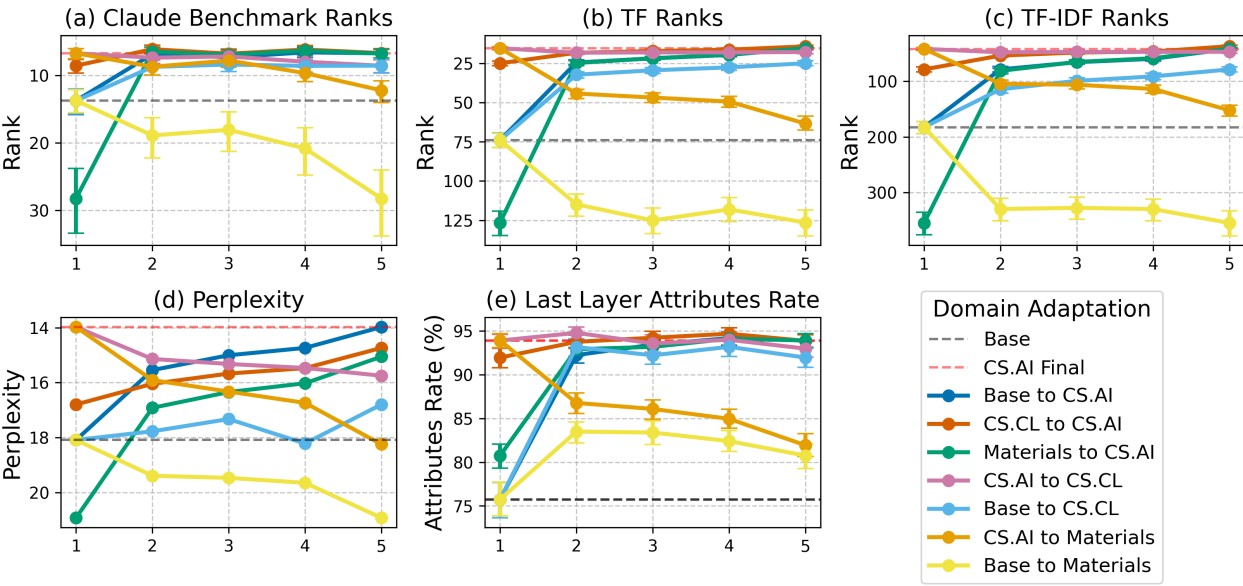

Figure 9: The x-axis shows training steps during domain adaptation, while the y-axis represents different evaluation metrics across five panels: (a) Claude Benchmark Ranks, (b) TF Ranks, (c) TF-IDF Ranks, (d) Perplexity, and (e) Last Layer Attributes Rate. Each colored line represents a different domain adaptation path, with the legend indicating the starting and ending domains. As can be seen, models ending in `CS.AI` (blue, red, and green lines) consistently achieve better ranks across all metrics, while models adapted to `Materials Science` (yellow and orange lines) show the worst performance. The TF and TF-IDF ranks closely follow the Claude benchmark trends, whereas perplexity does not give a fine grade distinction during adaptation."

## A.7   MMLU Experiments

This section demonstrates why current benchmark evaluation methods may not be optimal for model assessment. We investigated this by conducting experiments on the Computer Science subdomain (College and High School levels) from MMLU with a total of 200 questions of multiple-choice type (100 each). We employed three different approaches: first, we examined the most commonly used method for evaluating base models (few-shot learning) by introducing bias into the few-shot examples and observing the effects on model output. Second, we tested chat models to determine whether the bias toward correct options persists in instruction-tuned chat models. Third, we investigated potential bias in prompting strategies for chat models.

### A.7.1   Few-Shot Learning Bias

The objective of this experiment was to evaluate bias in the most common approach for benchmark evaluation of base models (Brown et al., 2020; Hendrycks et al., 2020), specifically, few-shot learning. To test the stability of accuracy metrics obtained through few-shot learning, we designed a controlled experiment.

We first evaluated MMLU using the standard methodology for the two Computer Science subdomains, utilizing the few-shot examples provided in each case (5 examples per subdomain in MMLU). We then systematically biased the options in all few-shot examples toward a single option (e.g., making Option A correct for all few-shot examples) while ensuring that this option was not the actual correct answer by randomly exchanging it with other options. This procedure was repeated for all answer options (A, B, C, and D). Additionally, we included a simple shuffling condition where options for few-shot examples were randomly shuffled along with their corresponding correct answers, repeating this process three times.

As shown in Supplementary Figure 11, smaller models exhibit significant performance changes due to this bias, though the effect persists even in larger models.

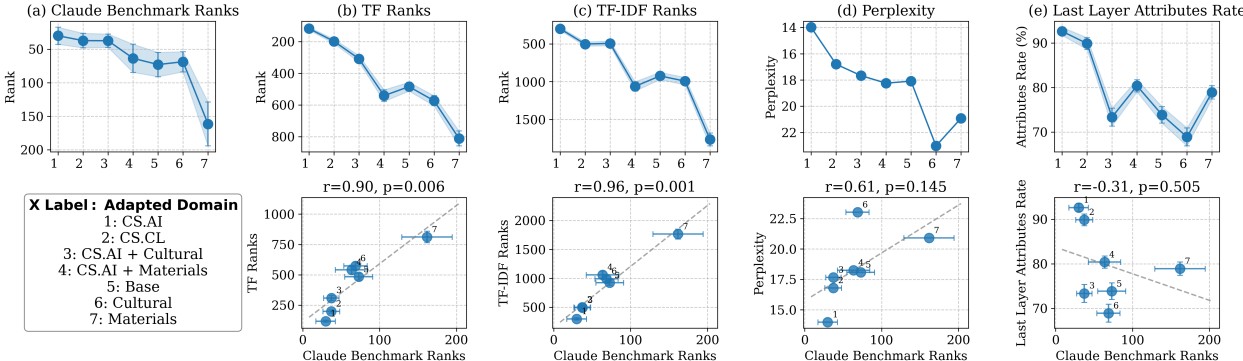

Figure 10: The x-axis shows adapted domains ordered by dissimilarity to `CS.AI`, while the y-axis represents evaluation metrics across five panels using mean aggregation. Top row: Prediction ranks on (a) the benchmark generated by Claude Sonnet 4, and (b-c) TF- and TF-IDF-based benchmarks produced by our pipeline, along with baseline metrics (d) perplexity and (e) last-layer attribution rate. Bottom row: Correlation analysis between each metric and the Claude benchmark. As can be seen, the mean aggregation produces higher rank values compared to Figure 8 due to the inclusion of outliers, but maintains the same overall pattern where models adapted to domains similar to `CS.AI` achieve better performance than those trained on unrelated domains like `Materials`. The correlations remain significant between the Claude benchmark and our TF-based methods, validating the robustness of our pipeline even when outliers are included in the evaluation.

### A.7.2 Option Bias in Chat Models

The second most popular benchmarking method involves using chat models instead of base models when available. To investigate whether option bias exists in these models, we shuffled the answer options three times and tested scenarios where all correct answers were either all A, B, C, or D.

As demonstrated in Supplementary Figure 12, bias persists significantly in chat models, with particularly pronounced effects in models such as Llama-2 7B Chat and OLMo 2 7B Instruct.

### A.7.3 Prompting Bias in Chat Models

These experiments demonstrate that bias extends beyond answer option ordering to include prompt structure effects. We tested different prompt variations as detailed in Supplementary Table 6. Supplementary Figure 13 reveals that even identical prompts written in different formats can significantly affect model accuracy, with relatively longer prompts generally yielding diminished performance.

### A.8 Smaller Language Models and Few-Shot Learning: An Example for GPT2-XL Model

To demonstrate the sensitivity to few-shot learning faced in smaller language models, we take one example for the GPT2-XL model. We tested GPT-2 XL on simple geography questions using greedy decoding (`do_sample=False`, `num_beams=1`).

When presented with a single question, GPT-2 XL demonstrated correct factual knowledge:

**Prompt 1**

```
Question: Which of the following is the capital of France?
A) Berlin
B) Madrid
C) Paris
D) Rome
```

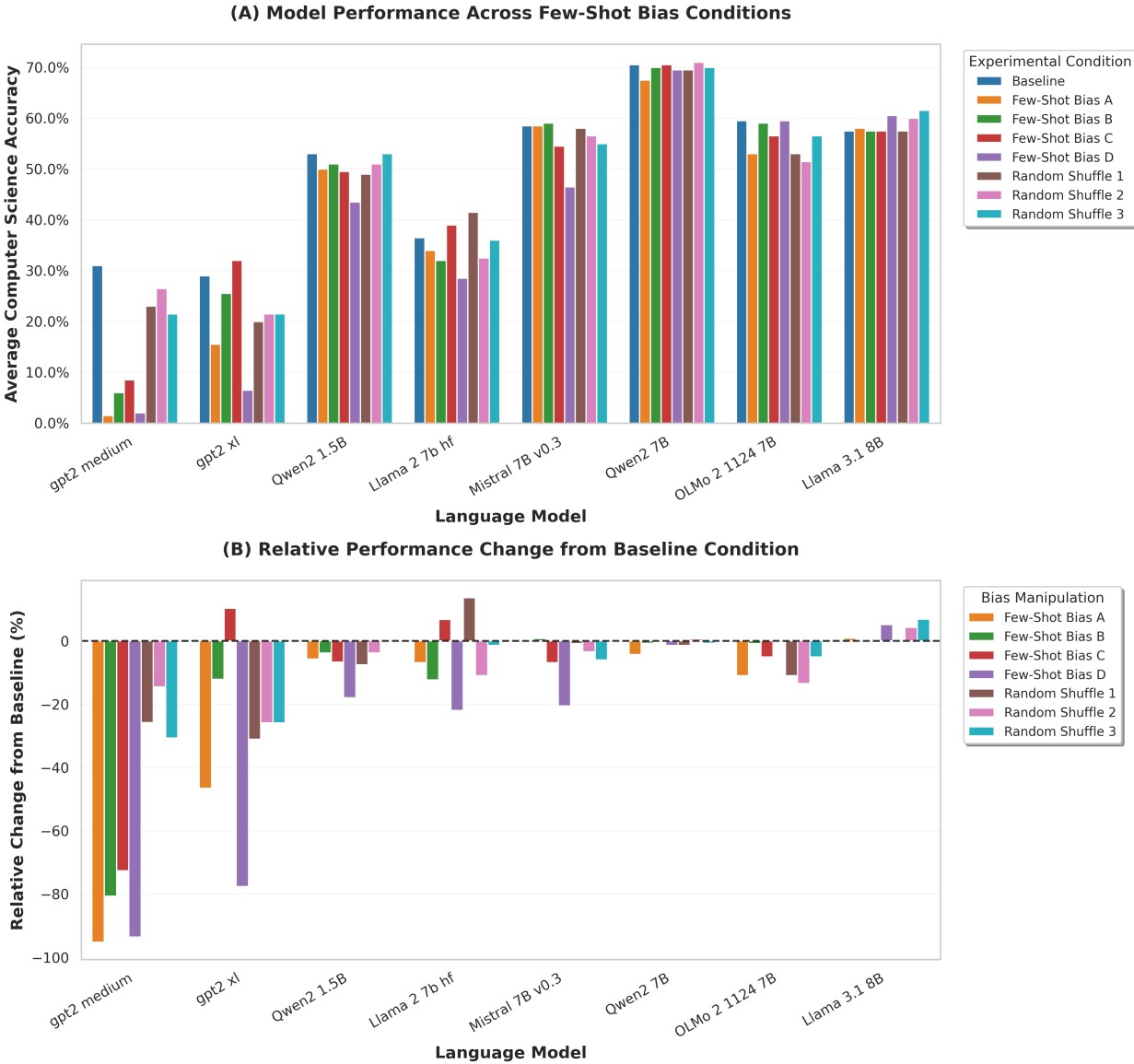

Figure 11: The x-axis shows different language models, while the y-axis shows average Computer Science accuracy (panel A) and relative performance change from baseline (panel B). Different colors represent various bias conditions: baseline, few-shot bias toward options A/B/C/D, and random shuffling variants. Smaller models exhibit more substantial performance degradation under biased conditions, though bias effects persist across all model sizes.

```
Answer:
```

For this prompt, GPT-2 XL correctly completed with "C", identifying Paris.

However, when presented with a sequence of questions in a few-shot format:

**Prompt 2**

```
Question: Which of the following is the capital of Germany?
A) Paris
```

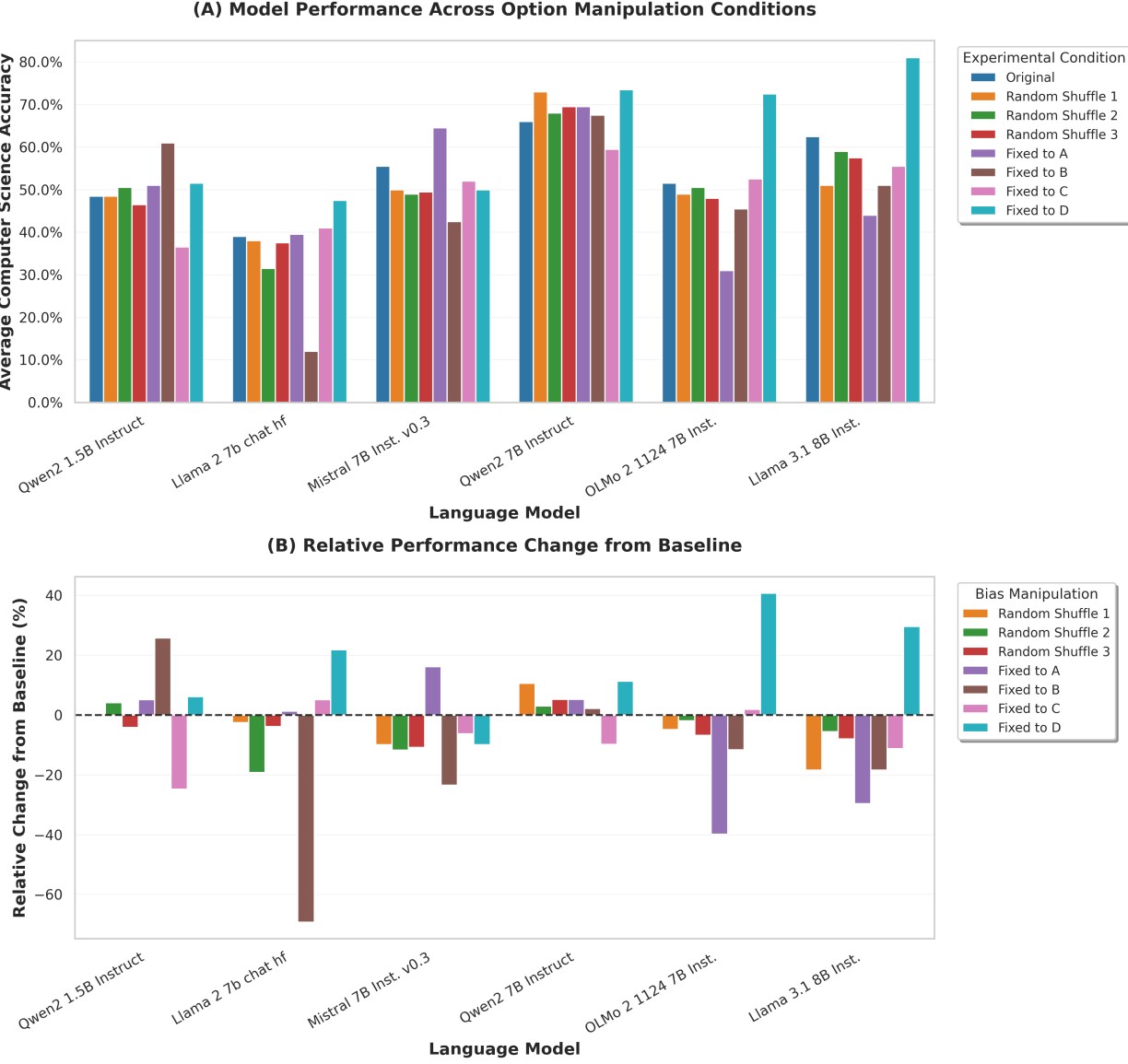

Figure 12: The x-axis shows different chat models, while the y-axis displays average Computer Science accuracy (panel A) and relative performance change from original conditions (panel B). Different colors represent various option manipulation conditions: original, random shuffling variants, and systematic bias toward options A/B/C/D. Option bias significantly affects chat model performance, with some models like Llama-2 7B Chat and OLMo 2 7B Instruct showing particularly strong susceptibility to option ordering effects.

```
        B) Berlin
        C) Warsaw
        D) Vienna

        Answer: B

        Question: Which of the following is the capital of Italy?
        A) Madrid
        B) Athens
```

**(A) Model Performance Across Prompting Style Variations**

**(B) Relative Performance Change from Standard Prompting**

Figure 13: The x-axis shows different chat models, while the y-axis displays average Computer Science accuracy (panel A) and relative performance change from standard prompting (panel B). Different colors represent various prompting styles: standard, minimal, detailed, casual tone, formal tone, and directive tone. Prompting variations substantially impact model performance, with detailed prompts generally reducing accuracy compared to standard approaches.

```
    C) Rome
    D) Lisbon

    Answer: C

    Question: Which of the following is the capital of Spain?
    A) Rome
    B) Paris
    C) London
```

Table 6: Prompt variations used to evaluate prompting bias in chat models

| STYLE | PROMPT TEMPLATE |
| --- | --- |
| Standard | *System:* "Answer the multiple choice question by providing only the letter (A, B, C, or D) of the correct answer."
*User:* "The following is a multiple choice question about {subject}.\n\n{question}" |
| Minimal | *System:* "Pick A, B, C, or D."
*User:* "{question}" |
| Detailed | *System:* "You are an expert academic assistant with deep knowledge across multiple disciplines. Your role is to analyze multiple choice questions with precision and accuracy. When presented with a question, carefully examine all provided options, consider the nuances of each choice, apply relevant domain knowledge, and select the single most appropriate answer. Always respond with only the letter (A, B, C, or D) that corresponds to your chosen option. Maintain objectivity and base your decisions on factual accuracy and logical reasoning."
*User:* "You are an expert in {subject}. Below is a comprehensive multiple choice question that requires careful analysis and reasoning. Please read the question thoroughly and consider all the provided options before making your decision.\n\nInstructions:\n- Analyze each option carefully\n- Consider the context and specific details mentioned in the question\n- Use your knowledge of {subject} to evaluate the correctness of each choice\n- Select the single best answer that most accurately addresses the question\n- Provide only the letter (A, B, C, or D) corresponding to your chosen answer\n\nQuestion and Options:\n{question}\n\nRemember to base your answer solely on your understanding of {subject} and the information provided in the question. Choose the most appropriate option and respond with only the letter of your choice." |
| Casual Tone | *System:* "Answer multiple choice questions by selecting the best option. Reply with just the letter."
*User:* "Here's a multiple choice question about {subject}. Please select the best answer.\n\n{question}" |
| Formal Tone | *System:* "Please choose the most suitable answer from the given choices and respond with the letter only."
*User:* "Please answer the following multiple choice question relating to {subject}. Select the most appropriate response.\n\n{question}" |
| Directive Tone | *System:* "Solve the multiple choice question by selecting the correct option. Provide only the letter."
*User:* "Solve this {subject} multiple choice question by choosing the correct answer.\n\n{question}" |

```
D) Madrid

Answer: D

Question: Which of the following is the capital of France?
A) Berlin
B) Madrid
C) Rome
```

Table 7: Prompt and corresponding target examples for given chapters from the `CS.AI` expert benchmark, drawn from *Understanding Deep Learning* (Prince, 2023).

| CHAPTER | PROMPT | TARGET |
|---|---|---|
| Regularization | We start by considering regularization in its strictest sense. Then we show how the stochastic gradient descent algorithm itself favors certain solutions. This is known as | implicit regularization |
| Regularization | A second approach is to generate several different datasets by re-sampling the training data with replacement and training a different model from each. This is known as | bootstrap aggregating |
| Residual Networks | Nearby gradients are correlated for shallow networks, but this correlation quickly drops to zero for deep networks. This is termed the | shattered gradients |
| Residual Networks | A series of these models form a stacked hourglass network that alternates between considering the image at local and global levels. Such networks are used for | pose estimation |
| Graph Neural Networks | By contrast, a transductive model considers both the labeled and unlabeled data at the same time. It does not produce a rule but merely a labeling for the unknown outputs. This is sometimes termed | semi-supervised learning |
| Graph Neural Networks | Unfortunately, if there are many layers and the graph is densely connected, every input node may be in the receptive field of every output, and this may not reduce the graph size at all. This is known as the | graph expansion problem |

```
    D) Paris

    Answer:
```

The model incorrectly completed with "B" (Madrid), despite being able to answer correctly when prompted without a few-shot prompt. This inconsistency highlights how smaller base models can sometimes fail to properly utilize in-context examples, reinforcing the need for our more structured approach to knowledge testing that reduces formatting sensitivity.

### A.9 Cosine Similarity Between Targets and Computer Science Phrase for GPT2-XL Adaptive Training

To measure the relevance of the targets predicted better by the `CS.AI` domain-trained model as compared to the base model. We developed a token-level model comparison metric that addresses several limitations in standard evaluation approaches. First, we computed both probability and rank differences between our `CS.AI` model and baseline models for each target token. To account for scale bias in probability differences and to balance the influence of both metrics, we created a composite score by normalizing both probability and rank improvements to equal scales and giving them equal weight (50% each). Recognizing that token frequency can distort analysis by over-emphasizing rare tokens with extreme differences, we applied a logarithmic frequency weighting to this composite score. The resulting composite weighted metric emphasizes improvements on commonly occurring tokens while maintaining sensitivity to meaningful differences in rare but potentially essential tokens. We employed the all-MiniLM-L6-v2 sentence transformer for semantic analysis to embed each token and the reference domain term "Computer Science Artificial Intelligence" into a shared vector space, calculating cosine similarity to quantify domain relevance. Statistical significance between the top and bottom performance groups was assessed using independent samples t-tests with Cohen's d for effect size quantification. Figure 15 shows a semantic alignment between `CS.AI`'s performance and specialized domain knowledge. Tokens, where `CS.AI` performed better (top 25% by composite_weighted score), showed

Table 8: Prompt and corresponding target for given keywords from `Computer Science - Artificial Intelligence (CS.AI)` and `Quantitative Biology - Populations and Evolution (Q-Bio.PE)` domains.

| KEYWORD | PROMPT | TARGET |
|---|---|---|
| **Domain: Computer Science - Artificial Intelligence (`CS.AI`)** | | |
| Machine Learning | One approach, which has recently been immensely successful in machine learning with large artificial neural networks, is the idea of learning through gradient descent using the | backpropagation |
| Machine Learning | Machine learning algorithms that learn, and utilize the causal relationship amongst variables provide better generalization performance, robustness against | adversarial |
| Machine Learning | Federated Learning is an advanced machine learning technique proposed by | Google |
| Reinforcement Learning | Prior attempts at improving data efficiency in reinforcement learning, involved the use of an Experience | Replay |
| Reinforcement Learning | It allows an agent to learn a policy to maximize a possibly delayed reward signal in a stochastic environment and guarantees convergence to an optimal policy, provided that the agent can sufficiently experiment and the environment in which it is operating is | Markovian |
| Reinforcement Learning | AIRL is an inverse reinforcement learning algorithm based on an | adversarial |
| Deep Learning | From a different perspective, Learning Important Features Through Propagating Activation Differences introduced | DeepLIFT |
| Deep Learning | The DBN is proposed by , it builds on multiple restricted | Boltzmann |
| Deep Learning | In particular, the workhorse of modern deep learning, the backpropagation algorithm, has proven difficult to translate to | neuromorphic |
| **Domain: Quantitative Biology - Populations and Evolution (`Q-Bio.PE`)** | | |
| Phylogenetic Tree | In 2004, Speyer and Sturmfels showed a space of phylogenetic trees with a given set of labels on their leaves is a | tropical |
| Phylogenetic Tree | The eventual goal of our work is to provide a divide-and-conquer strategy for Bayesian phylogenetics, in which taxa are divided into subsets, a Bayesian analysis is run on each, and then knitted back together using a | supertree |
| Phylogenetic Tree | The phylogenetic tree and the proteins used are those which Adachi and Hasegawa used to estimate mtREV; the Japanese | mtDNA |
| Disease Spread | This finding is consistent with the theory of infectious disease spread in highly coupled | metapopulations |
| Disease Spread | Another important parameter that wholly describes the spread of an outbreak is the basic reproduction number $(R_0)$, that is computed as the ratio between the transmission rate and the sum of recovery and specific | mortality |
| Disease Spread | A branching random-walk model of disease outbreaks and the | percolation |
| Evolutionary Dynamics | In the context of evolutionary dynamics, much of the quantitative work has been strongly inspired by two famous analogies: Fisher's tentative link between natural selection and the second law of | thermodynamics |
| Evolutionary Dynamics | Quasispecies dynamics with network constraints : A quasispecies is a set of interrelated | genotypes |
| Evolutionary Dynamics | Our result connects game theory to the core of evolution theory, by providing an unusual point of view: Evolution is a coordination game between | genes |

---

**Benchmark Creation Template**

```
# Domain-Specific Benchmark Creation Instructions
Generate exactly 1000 prompt-target pairs for **{domain name}** domain:
50 prompts per category across 20 categories.

## Categories
{list of 20 categories}

## Key Rules
1. **Prompt Structure**: Must end with complete phrases including any
   needed articles/stopwords:
   - "is called a", "is known as an", "is termed the", "are called",
     "refers to a", etc.
   - 15-40 words total, easy-to-medium difficulty
2. **Target Structure**:
   - Starts with space: `" target_term"`
   - NO stopwords (a, an, the, etc.) in target
   - Must belong to the category (not necessarily be the category name)
3. **Quality**: Unambiguous, diverse structures, comprehensive coverage
   per category

## Examples
**Correct**:

qa_pairs = [
# Category 1: {category_1_name}
{"prompt": "The entity that takes actions in an environment to maximize
 reward is known as an", "target": " agent"}
{"prompt": "Software that learns optimal behavior through interaction
 is called a", "target": " learner"}
...]
qa_pairs.extend([
# Category 2: {category_2_name}
{"prompt": "Methods that optimize policies without environment models
 are termed", "target": " model-free algorithms"}
....])
```

Figure 14: **Template:** Complete prompt template provided to Claude Sonnet 4 for domain-specific benchmark creation.

significantly higher semantic similarity to "Computer Science Artificial Intelligence" compared to tokens with the lowest performance (p = 2.82e-05, Cohen's d = 0.31). Domain-specific terms like " classifiers," " algorithms," and "program" clustered in the high-performance group, while semantically distant terms like " fMRI" and " boxes" appeared mainly in the low-performance group. The violin distributions reveal different central tendencies and variance patterns between groups. Importantly, token frequency (shown by point size) was distributed across both similarity ranges, confirming our weighting approach successfully prevented frequency bias from dominating the analysis. Complete lists of the top and bottom 100 tokens further support this finding (Table 9 and Figure 15), with specialized AI terminology concentrated in the high-performance group. These results provide strong evidence that CS.AI has developed genuine domain expertise rather than merely statistical advantages, validating results for our benchmark.

Table 9: Tokens with highest and lowest performance improvements in the CS.AI domain-trained model

| Top 100 Tokens | Bottom 100 Tokens |
|---|---|
| Policy, Answer, tracing, margin, bases, policy, actions, baselines, answer, answering, Critic, action, planner, reward, Error, prediction, agent, explainability, MDP, generalization, weakly, critic, graphs, proposed, contrastive, efficiency, paper, Interpretable, Diversity, exploration, partially, approximate, Recall, Logic, learnt, Monte, Support, Atari, labeled, continual, Search, Partially, training, steps, segmentation, Inverse, Mixed, NMT, distillation, detection, Markov, reinforcement, best, interacting, states, classifiers, adaptation, loss, segment, UAV, Context, homogeneous, Explainable, policies, federated, demonstration, discriminator, language, Freebase, commonsense, unsupervised, control, rewards, PSO, diarization, tabular, differentiable, Table, privacy, entities, made, regularization, intersection, complexity, agents, publicly, mentions, natural, matching, spaces, relations, catastrophic, research, intrusion, program, SNNs, systems, DDPG, future, confident | parsing, one, like, discuss, Multivariate, Tree, destination, understanding, autoencoders, sensor, outliers, squared, offensive, structures, percent, languages, MRI, drones, science, score, video, fields, explainers, public, ODE, credit, SGD, may, decoder, vertices, logistic, contains, transitions, problem, Generation, backpropagate, drive, output, attacks, world, allocating, trials, sources, magnetic, size, implementation, field, signals, behavior, analysis, protect, cards, nonlinear, Absolute, Science, way, PDEs, strategy, land, emotional, percentage, French, improved, dimensionality, classical, observation, hyperparameters, Facebook, percents, private, example, theoretical, syntactic, domain, Code, bound, analogy, MCMC, explainer, truth, multivariate, layer, mergers, PDE, protection, fMRI, multivariable, imaging, media, direction, allocate, MRIs, Anomaly, government, boxes, ratio, equations, FMRI, chart, differential |

Note: Tokens are listed in descending order of performance improvement as measured by the composite weighted metric. The top tokens show significantly higher semantic similarity to the CS.AI domain ($p = 2.82e{-}05$, Cohen's d = 0.31).

### A.10 Computational Budget

Our experiments utilized both H100 and A100 GPU infrastructure with a total computational budget of approximately 1000 CPU hours and 1300 GPU hours. The Llama model adaptive and continual learning experiments were conducted on H100 GPUs, requiring approximately 800 hours of training time. The remaining GPU computations, including GPT-2 XL domain adaptation and model evaluations, were performed on A100 GPUs for the remaining approximately 400 hours. Most components of our benchmark construction pipeline, including keyword extraction, sentence matching, and prompt-target pair generation, rely primarily on CPU processing with only minimal GPU usage and represent the majority of the 1000 CPU hours. All experiments were conducted using standard deep learning frameworks with mixed precision training where applicable to optimize computational efficiency.

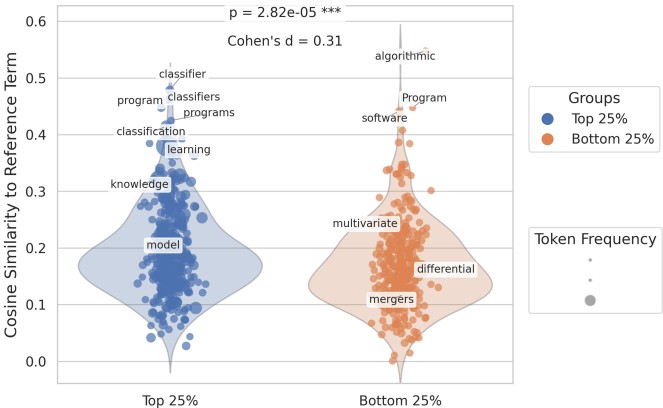

Figure 15: The x-axis shows two groups (Top 25% and Bottom 25% token performance), while the y-axis represents cosine similarity to the reference term "Computer Science Artificial Intelligence." The violin plot displays the distribution of semantic similarities for tokens where the `CS.AI` model performed best versus worst compared to the base GPT-2 XL model. As can be seen, tokens in the top 25% group show significantly higher semantic similarity to the CS.AI domain, with domain-specific terms like "classifiers," "model," and "program" appearing in the high-similarity region, while semantically distant terms like "mergers" and "differential" cluster in the low-similarity region. Point sizes indicate token frequency, demonstrating that the frequency weighting successfully prevented bias from dominating the analysis.

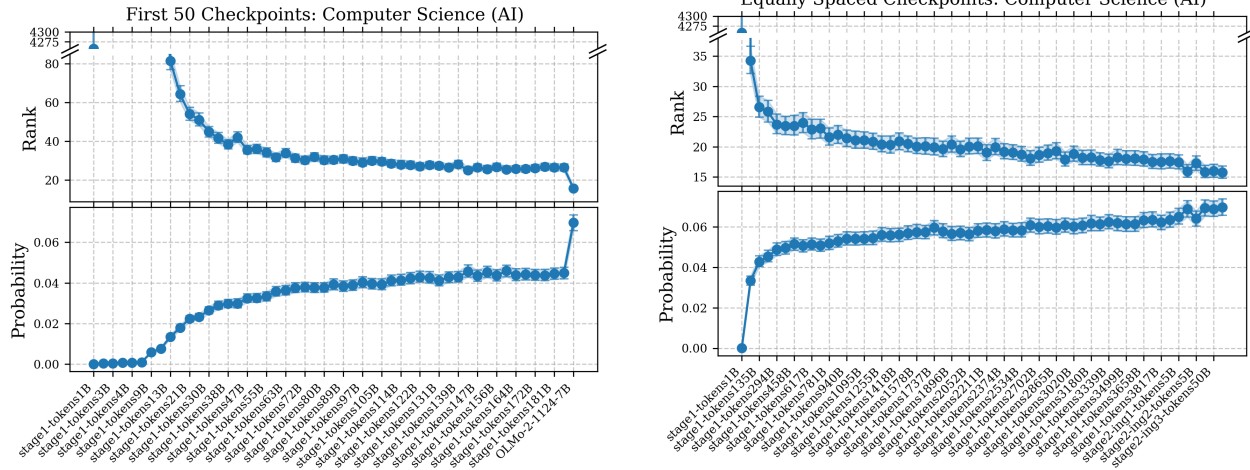

Figure 16: Prediction ranks and probabilities of OLMo-2 pretraining checkpoints on the `Computer Science (AI)` domain. The left panel displays results across the first 50 checkpoints, while the right panel shows equally spaced checkpoints from the entire pretraining, with the final data point representing the completed model. The x-axis shows checkpoint progression during training, while the y-axis represents ranks (top) and probabilities (bottom). As can be seen, the patterns mirror those observed in Fig. 5 for the `Physics (Soc-Ph)` domain, but with relatively lower ranks overall since the model demonstrates stronger baseline knowledge in the `CS.AI` domain compared to physics, reflecting the abundance of computer science training data in the pretraining corpus.

Table 10: OLMo-2 checkpoint models used in our analysis

| Category | Checkpoint Details | Token Range |
|---|---|---|
| First 50 | **OLMo-2-1124-7B__checkpoint-stage1-step** 150(1B), 600(3B), 700(3B), 850(4B), 900(4B), 1000(5B), 2000(9B), 2150(10B), 3000(13B), 4000(17B), 5000(21B), 6000(26B), 7000(30B), 8000(34B), 9000(38B), 10000(42B), 11000(47B), 12000(51B), 13000(55B), 14000(59B), 15000(63B), 16000(68B), 17000(72B), 18000(76B), 19000(80B), 20000(84B), 21000(89B), 22000(93B), 23000(97B), 24000(101B), 25000(105B), 26000(110B), 27000(114B), 28000(118B), 29000(122B), 30000(126B), 31000(131B), 32000(135B), 33000(139B), 34000(143B), 35000(147B), 36000(151B), 37000(156B), 38000(160B), 39000(164B), 40000(168B), 41000(172B), 42000(177B), 43000(181B), 44000(185B) | 1B - 185B |
| Equally Spaced | **OLMo-2-1124-7B__checkpoint-stage1-step** 150(1B), 13000(55B), 32000(135B), 51000(214B), 70000(294B), 89000(374B), 109000(458B), 128000(537B), 147000(617B), 166000(697B), 186000(781B), 205000(860B), 224000(940B), 242000(1016B), 261000(1095B), 280000(1175B), 299000(1255B), 319000(1338B), 338000(1418B), 357000(1498B), 376000(1578B), 395000(1657B), 414000(1737B), 433000(1817B), 452000(1896B), 470000(1972B), 489000(2052B), 508000(2131B), 527000(2211B), 547000(2295B), 566000(2374B), 585000(2454B), 604000(2534B), 624000(2618B), 644000(2702B), 663000(2781B), 683000(2865B), 701000(2941B), 720000(3020B), 739000(3100B), 758000(3180B), 777000(3259B), 796000(3339B), 815000(3419B), 834000(3499B), 853000(3578B), 872000(3658B), 891000(3738B), 910000(3817B), 928646(3896B) **+ Stage2-ingredient** checkpoints: ingredient1-step1000(5B), ingredient1-step11931(50B), ingredient2-step1000(5B), ingredient2-step11931(50B), ingredient3-step11931(50B) | 1B - 3896B |
| Final Model | OLMo-2-1124-7B | Complete |

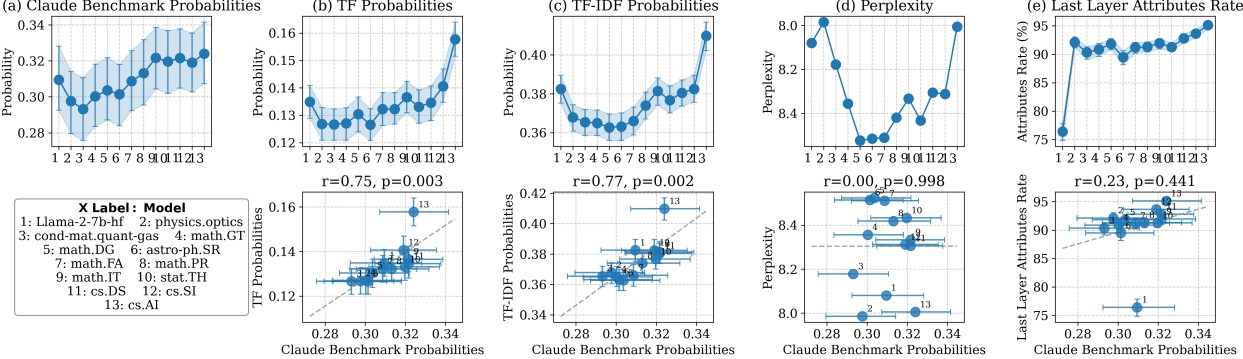

Figure 17: Probability values for 12 checkpoints from continual pretraining of Llama2-7B, along with the base model. The bottom-left panel displays the sequence of pretraining domains in the order they were introduced during training. Top row: Target token probabilities on *(a)* the benchmark generated by Claude Sonnet 4, and *(b–c)* TF- and TF-IDF-based benchmarks produced by our pipeline. The last two columns show the baseline metrics *(d)* perplexity and *(e)* last-layer attribution rate are the same as the main Fig. 6 but kept for comparison purposes. Bottom row: Correlation of each metric with the Claude benchmark. Compared to the rank-based analysis in Fig. 6, probability values provide a normalized view of knowledge acquisition dynamics, as it is limited between 0 and 1, revealing subtle changes in model performance that may be obscured when using rank aggregations. The probability trends similarly demonstrate that our TF- and TF-IDF-based metrics capture meaningful knowledge changes aligned with domain similarity to `CS.AI`, while traditional metrics like perplexity fail to provide reliable signals for continual pretraining progress.

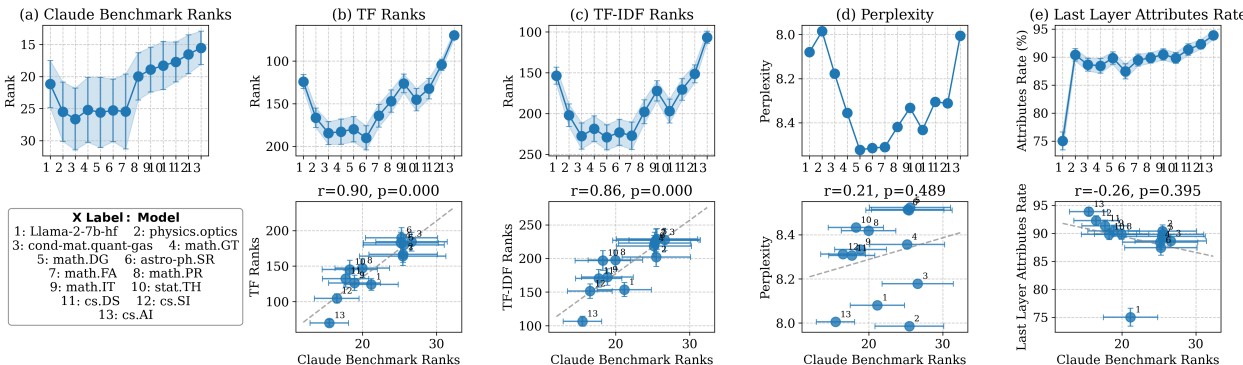

Figure 18: Mean rank aggregation for 12 checkpoints from continual pretraining of Llama2-7B, corresponding to Fig. 6. Unlike the trimmed mean (20%) aggregation, this analysis includes potential outliers, resulting in larger y-axis values but smoother trends. The mean aggregation reveals even stronger correlations between our TF- and TF-IDF-based metrics and the Claude benchmark (panels b-c), demonstrating that the inclusion of all data points provides more nuanced insights into knowledge acquisition dynamics during continual pretraining.

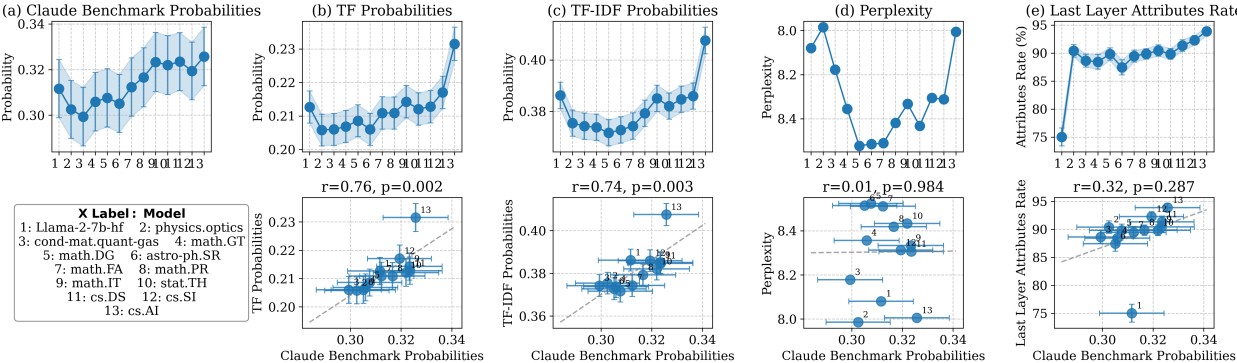

Figure 19: Mean probability aggregation for 12 checkpoints from continual pretraining of Llama2-7B, corresponding to the probability analysis in Fig. 17. Similar to the mean rank analysis, this aggregation includes all data points without trimming outliers, maintaining the same trends observed in the trimmed mean analysis while providing a complete view of the probability distributions across all generated prompts.

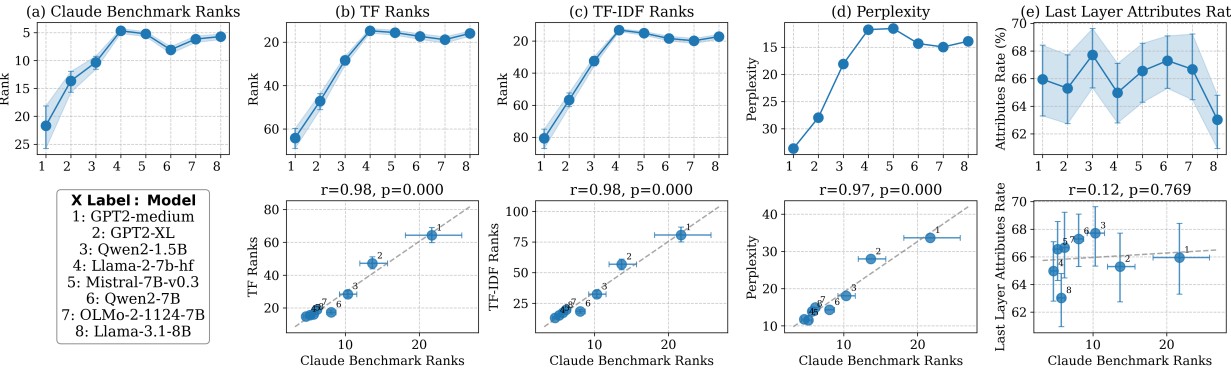

Figure 20: Evaluation of 8 base models on `General Economics (Econ-GN)` domain, including GPT-2 variants to demonstrate pattern consistency across model scales. The top row shows prediction ranks from (a) Claude benchmark, (b) TF-based method, (c) TF-IDF-based method, (d) perplexity, and (e) last layer attribution rates. The bottom row shows correlations between Claude benchmark ranks and each metric, demonstrating strong correlations for TF- and TF-IDF-based methods while perplexity shows high correlation and attribution rates show weak correlation.

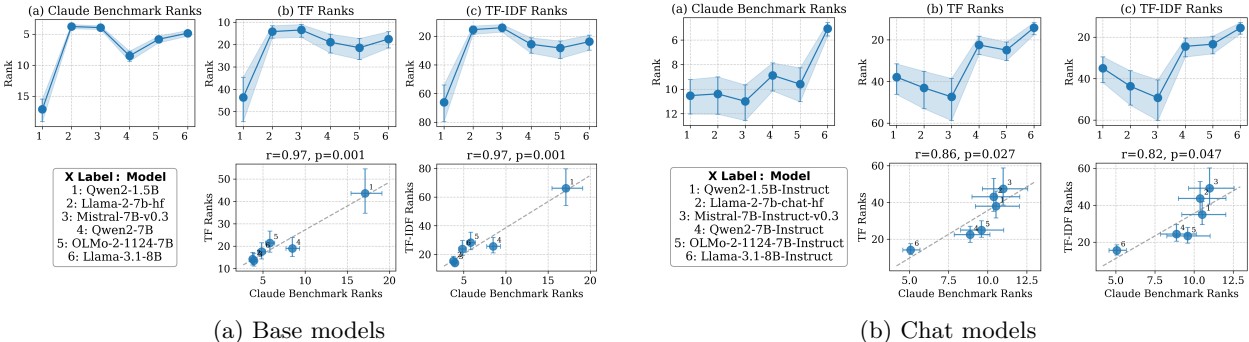

Figure 21: Evaluation of 6 base models (left) and 6 chat models (right) on `Quantitative Biology - Populations and Evolution (Q-Bio.PE)` domain. The top row shows prediction ranks and the bottom row shows the correlations between the Claude benchmark and our TF- and TF-IDF-based benchmarks.

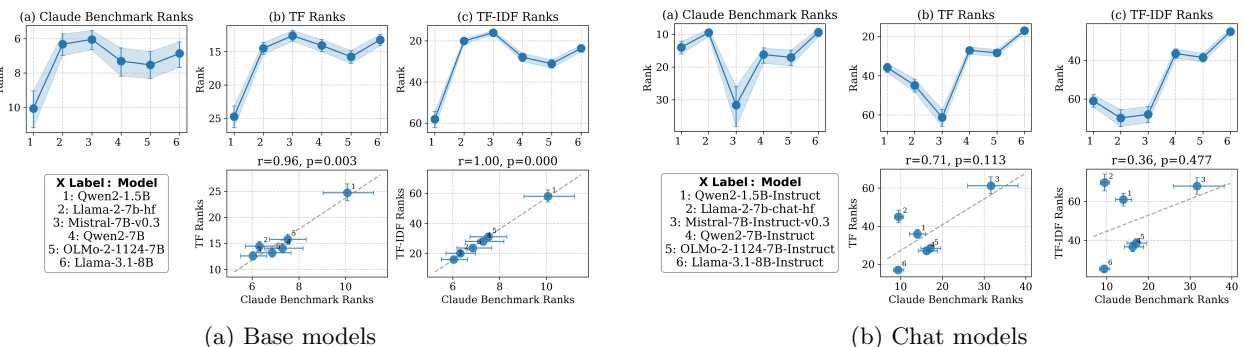

Figure 22: Evaluation of 6 base models (left) and 6 chat models (right) on `CS.AI` domain. The top row shows prediction ranks and the bottom row shows the correlations between the Claude benchmark and our TF- and TF-IDF-based benchmarks.

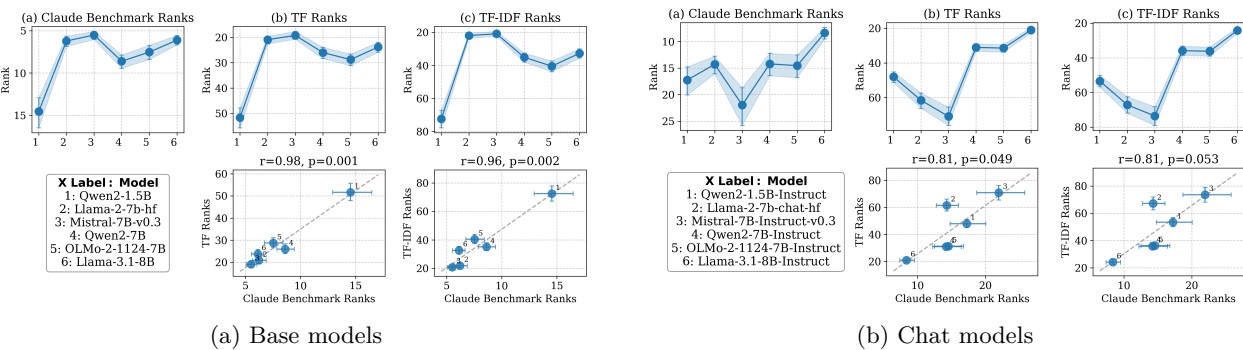

Figure 23: Evaluation of 6 base models (left) and 6 chat models (right) on `Physics (Soc-Ph)` domain. The top row shows prediction ranks and the bottom row shows the correlations between the Claude benchmark and our TF- and TF-IDF-based benchmarks.

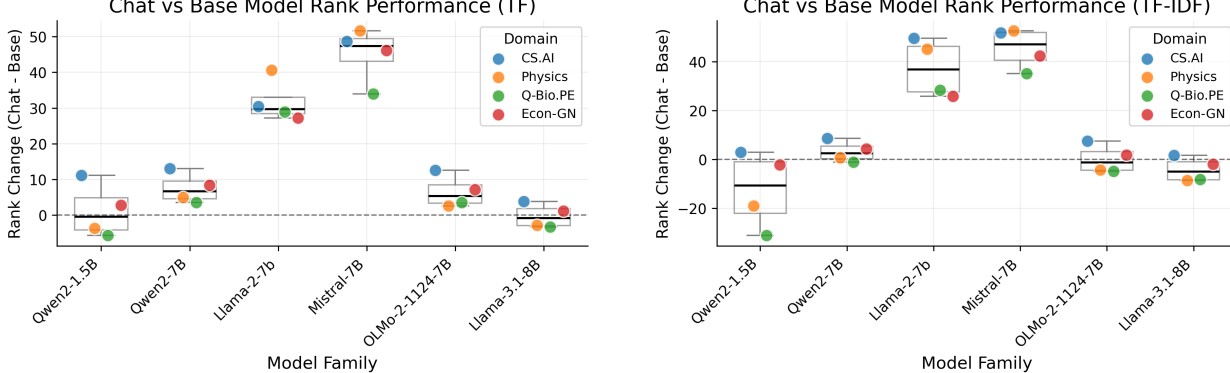

Figure 24: Impact of instruction tuning on model performance across domains, showing the difference between trimmed mean ranks (Chat - Base) for TF-based (left) and TF-IDF-based (right) evaluation methods. Positive values indicate that chat models perform worse than their base counterparts, while negative values indicate improvement. Both evaluation methods show consistent patterns: for most model families, base models generally outperform their chat-aligned counterparts across domains. The degradation is particularly pronounced for Llama-2-7B and Mistral-7B-v0.3, which show substantial performance drops after instruction tuning across `CS.AI`, `Physics (Soc-Ph)`, `Q-Bio.PE`, and `Econ-GN` domains, suggesting significant room for improvement in their alignment pipelines.

