# OpenReview forum: "From Raw Corpora to Domain Benchmarks: Automated Evaluation of LLM Domain Expertise"
_TMLR — Rejected by TMLR_

### Review · Reviewer_ZQH5 · 2026-04-26

**Summary Of Contributions:**

This paper proposes a deterministic pipeline that transforms raw domain corpora into completion-style benchmarks for evaluating LLM domain expertise. The pipeline extracts domain-specific keywords via n-gram  construction and embedding-based filtering, builds TF and TF-IDF target vocabularies, and constructs prompt-target pairs where domain-specific terms serve as prediction targets. Models are evaluated by the rank of  the correct target token in their output distribution. The authors validate against an expert-curated benchmark (r=0.99) and a Claude-generated proxy, then demonstrate the pipeline's utility in domain adaptation,  continual pretraining, general pretraining tracking, and base-vs-chat model comparison across four STEM domains.

Strengths:
1. This paper addresses an important problem with strong practical value: building an automated pipeline to evaluate LLM performance in specific domains.
2. The core idea is straightforward and logical, with a useful dynamic property that naturally prevents data contamination issues.

Weaknesses:
1. The domain-specific evaluation constructed in this paper focuses on word-level metrics, which can only assess familiarity with domain vocabulary rather than the ability to handle domain-specific reasoning and problem-solving. Could the authors discuss this limitation and potential improvements?
2. The method relies heavily on the performance of the LLMs used in the pipeline. Although top closed-source LLMs currently have strong general capabilities, they may still produce significant errors in specialized domains. Cross-validating with experts and LLMs on only 6 samples is insufficient to demonstrate the method's effectiveness and generalizability.
3. The benchmark domains tested are relatively narrow, all are STEM-related and sourced from RedPajama. The generalizability of the method to other domains remains uncertain.

**Audience:**

Yes

**Audience Explanation:**

This paper addresses an important problem with strong practical value.

**Claims And Evidence:**

No

**Claims Explanation:**

The method relies heavily on the performance of the LLMs used in the pipeline and the benchmark domains tested are relatively narrow.

**Requested Changes:**

1. Could you elaborate on the evaluation capability of this method beyond word-level metrics?
2. Do you have more thorough cross-validation results, such as validation across more domains with a larger set of expert-annotated samples? Alternatively, could you address the concern that the proposed method relies too heavily on the capabilities of specific LLMs?
3. Do you have additional test results on a broader range of domains?

---

> ### Author Response · Authors · 2026-05-16
> **our response**
>
> We appreciate the reviewer's recognition of the practical value and the contamination-resistant property of our pipeline.
>
> **Comment 1 (word-level metrics).** Our framework is best understood as a cloze-style test of domain expertise. While some prompts incidentally require reasoning, most corpus-derived completions do not, and we agree that going beyond target-level prediction to assess reasoning and problem-solving is an exciting direction. We see our work as analogous to the role MCQ-style benchmarks played in the early days of LLM evaluation, which is a useful first step toward more comprehensive domain-aware evaluation, and we will discuss this framing explicitly in the revised limitations section.
>
> **Comment 2 (reliance on specific LLMs).** We would like to clarify that our pipeline itself uses **no LLMs** other than a small sentence-embedding model (`all-MiniLM-L6-v2`) for keyword–sentence matching. Keyword extraction, target vocabulary construction, and prompt–target pair generation are entirely deterministic. Claude is used **only** in the validation step, as an intermediate proxy between our pipeline and the (expensive) human expert benchmark. Crucially, we have now confirmed in two domains (CS.AI and Econ-GN; see the general response) that ranks computed on our pipeline correlate near-perfectly with those on independently curated human expert benchmarks, which do not involve Claude at all. This addresses the concern about over-reliance on any single LLM's capabilities.
>
> **Comment 3 (broader domains).** This is a fair point. Our four domains are all sourced from arXiv (RedPajama), with the partial exception of General Economics, which has notably different writing conventions. Adapting to non-arXiv corpora (e.g., legal or clinical text) would require domain-appropriate data cleaning. We emphasize, however, that **all pipeline hyperparameters were fixed on CS.AI** and applied unchanged to the other three domains, with consistent results across all of them — including the stylistically distinct Econ-GN domain. We take this as encouraging evidence that the pipeline generalizes, and we will state this caveat clearly in the limitations.

---

### Review · Reviewer_toFC · 2026-04-27

**Summary Of Contributions:**

**[Summary]**

This paper proposes an evaluation framework for assessing domain knowledge in large language models. In response to the unreliability of prior perplexity-based and MCQ-based evaluations, the paper introduces an automated pipeline that does not require expensive additional LLM annotation or human annotation. The pipeline first extracts domain keywords from the input corpus, then matches relevant sentences from domain full-text documents. Based on TF/TF-IDF target vocabularies, it constructs prompt–target pairs as a completion-based benchmark. The goal is to evaluate a model’s domain knowledge by measuring the ranking of the target token predicted by the model given a prefix prompt.
The annotation reliability and evaluation validity of this method are supported by a three-level validation system. The authors also demonstrate the generalizability of the method to other domains in the main text and the appendix. Experiments show that, for small and medium-scale models, the proposed method can verify the effects of knowledge transfer after domain adaptation and during domain continual pretraining, outperforming baselines such as perplexity and final-layer attribution.

**[Key Strengths]**
1. The method is cost-effective and highly deterministic. The pipeline does not rely on additional generative model annotation. It can be scaled to any raw corpora. This is helpful when developing models, as it can be used to exclude specific datasets and thereby reduce dataset contamination risks.
2. The experimental settings for analysis are comprehensive. Under controlled continued training of models, the paper examines multiple dimensions, including domain adaptation, tracking during pretraining, and comparisons across model types. These experiments show that the proposed method performs better than traditional baselines in quantifying the fluency, or familiarity, with which models use domain-specific vocabulary (Please see my comments below for why I use this wording).

**[Key Weaknesses]**
1. I have some doubts about whether the proposed method can really evaluate domain knowledge. First, the completion-based evaluation format is close to the alignment objective of next-token prediction in language modeling. It seems more likely to reflect the model’s “familiarity” with, or proficiency in, domain-related vocabulary, rather than necessarily extending to the ability to complete domain-related tasks in practical settings.
2. The method may be useful in practice for open-source models, but it seems difficult to extend to closed-source models. The authors rely on rankings over the full vocabulary, while most closed-source models, when accessed through APIs, only return top-K high-probability token completions. Although this could be regarded as an approximate substitute for ranking, it may introduce imprecision.
3. The construction of the Claude benchmark seems to involve some bias. The current validation benchmark is still built based on the top-k keywords first extracted by the proposed pipeline. This may not fully decouple the validation benchmark from the method being evaluated.

**Audience:**

Yes

**Audience Explanation:**

1. Evaluating domain capability indeed requires unbiased and scalable methods. In fact, many current benchmarks are also moving away from MCQ-style evaluation and instead focus on reasoning in realistic answer-generation settings (e.g., mathematical reasoning), or on capabilities such as tool use in agentic scenarios. Although the completion-based pipeline is simple, it can to some extent quantify familiarity with domain-related vocabulary.
2. The analysis part provide useful insights and reference value through comparisons across different models. Beyond reporting benchmark results, the paper studies domain-adapted models, the pretraining process, and capability transfer after instruction-following SFT. The conflicts with traditional metrics suggest that token rank may serve as a monitoring signal for model training dynamics.

**Broader Impact Concerns:**

The paper does not include a Broader Impact section. It may be useful to briefly discuss the method’s potential role in preventing data contamination and protecting data copyright.

**Claims And Evidence:**

No

**Claims Explanation:**

1. The authors may overgeneralize the scope of the claim about evaluating domain knowledge. I do not deny that the proposed method is a good proxy metric for quantifying a model’s familiarity with (or proficiency in) domain corpora. However, evaluating domain knowledge in most cases still depends on how well the model performs on concrete domain-specific tasks under realistic instructions. It is possible that the model only learns collocational patterns around domain terminology, without being able to carry out task reasoning.
2. The authors mention that the proposed method could be designed to avoid data contamination. It is true that when using this method, one can control the composition of the evaluation data. However, the experiments in the paper do not seem to clearly support this claim. For the RedPajama dataset used in this paper, the authors do not provide detailed discussion on whether there is any temporal overlap with the pretraining data of the evaluated models.
3. In Section 2.6, when the target consists of multiple tokens, the proposed sequential evaluation method needs to be augmented with the actual target tokens. This introduces bias when computing probabilities and rankings. Similar to the bias in teacher forcing, using the actual tokens does not perfectly measure the joint probability of the LM’s multi-step generation.

**Requested Changes:**

1. The claims in the paper regarding the evaluation of domain knowledge should be revised. It may be more appropriate to state that the proposed framework measures a model’s familiarity with domain texts, or that it has the potential to serve as a proxy monitoring metric during domain-transfer training.
2. Please explain how this method can be applied to closed-source models, or whether an approximate top-k ranking can substitute for ranking over the full vocabulary.
3. The Claude benchmark should be generated entirely using Claude.
4. Please report the knowledge cutoff dates of the evaluated models, or the dates of the training data used, as well as their overlap with the RedPajama dataset.
5. Please explain the possible teacher-forcing bias introduced by the sequential evaluation method, or consider using a different method to evaluate multi-token cases.
6. Presentation issues:
- Figure 1 is not mentioned in the main text (although it is referenced in Figure 2).
- Page 2: an Experience”-“replay”. -> missing “
- Page 6: probabilities in the Appendix; -> Please make it refer to specific section in Appendix.

---

> ### Author Response · Authors · 2026-05-16
> **our response**
>
> We thank the reviewer for the careful and detailed engagement with our work, and especially for the recognition of our analysis as comprehensive and our pipeline as cost-effective and scalable.
>
> **Comment 1 (familiarity vs. domain knowledge).** Please see the general response for our revised framing. In short, we agree that "familiarity with domain text" is the more cautious description, and we will adapt our terminology accordingly. We do, however, note that the near-perfect correlation with two independently curated human expert benchmarks (CS.AI and now Econ-GN) suggests the measure is picking up something more than corpus-specific surface familiarity.
>
> **Comment 2 (closed-source models).** This is a fair limitation. Our method requires access to rank or probability over the full vocabulary, which is not exposed by most closed-source APIs. We agree that approximating ranks via top-$k$ logprobs (where available) is a reasonable substitute but introduces imprecision, particularly for the long tail of low-probability completions where ranks may be uninformative anyway. We will add a discussion of this in the revised limitations section and frame full-vocabulary access as a methodological requirement rather than a fundamental constraint on the approach.
>
> **Comment 3 (Claude benchmark coupling).** Thank you for raising this; we welcome the opportunity to clarify our reasoning. In our initial experiments, Claude generated the benchmark end-to-end, proposing both the keywords and the prompt–target pairs, and we observed strong correlations with our pipeline ($p<0.001$). We subsequently revised the protocol to supply Claude with our extracted top-20 keywords, motivated by the view that **the keyword set is precisely the artefact a domain expert would be expected to curate or verify in practice**. Conditioning both methods on the same keywords enables a controlled comparison over an identical conceptual scope, isolating the contribution of the **prompt–target generation strategy itself** (corpus-derived vs. LLM-generated). Importantly, the prompts and targets produced by Claude remain independent of our pipeline's TF/TF-IDF target vocabulary and sentence-matching procedure, so the validation signal is not circular. We will make this design rationale explicit in the revision.
>
> **Comment 4 (knowledge cutoffs and RedPajama overlap).** Thank you for asking. We will include this in the revised paper. Known training data and approximate knowledge cutoffs of the evaluated models:
> | Model | Cutoff | Notes |
>  |---|---|---|
> | GPT-2 | ~early 2019 | WebText (Reddit-linked pages, scraped ~2017–2019) |
>  | Llama-2-7B | Sept 2022 | Mixture not fully disclosed |
> | Mistral-7B-v0.3 | ~early 2023 | Unofficial |
>  | Qwen2-1.5B / Qwen2-7B | ~late 2023 | Unofficial |
> | Llama-3.1-8B | Dec 2023 | Mixture not fully disclosed |
>  | OLMo-2-1124-7B | ~early 2024 | Training data public (Dolma) |
>
> Only GPT-2 (WebText) and OLMo-2 (Dolma) have publicly known training corpora. There is no published direct overlap measurement with RedPajama. However, both Dolma and RedPajama include arXiv, Wikipedia, and large subsets of Common Crawl (CC), so substantial overlap is plausible despite different deduplication and filtering pipelines. WebText, being URLs linked in Reddit, likely overlaps indirectly with CC-based subsets of RedPajama. We will add this discussion and underscore that, as Reviewer smLp also notes, our paper's contamination *resistance* claim should be framed as a property of future deployments on fresh corpora rather than a demonstrated property of the current experiments (please see also our reply to Reviewer smLp, Comment 4).
>
> **Comment 5 (teacher-forcing bias in multi-token targets).** This is a thoughtful concern. We chose sequential teacher-forced scoring because the standard alternative (letting the model freely generate and then string-matching against the target)  has a failure mode: if the model's first generated token is incorrect, no rank or probability can be computed for the *correct* continuation, mixing "wrong but reasonable alternative" with "no domain knowledge." Teacher-forced sequential scoring is essentially likelihood scoring of a cloze task, which is the standard practice for evaluating model knowledge in fill-in-the-blank settings. We acknowledge it introduces a bias toward measuring conditional knowledge rather than full joint generation; this is shared with all cloze-style evaluation. String-matching alternatives (LAMBADA-style) are also imperfect because corpus-derived targets often contain shorthand or compound forms (e.g., "teacher-forcing", "reinforcement learning (RL)") that models tokenise inconsistently. We will discuss this trade-off and the associated bias explicitly in the revised methods section.

---

### Review · Reviewer_smLp · 2026-05-09

**Summary Of Contributions:**

This paper proposes a deterministic pipeline for turning raw domain corpora into completion-style benchmarks for measuring LLM domain expertise. The pipeline extracts domain keywords, matches sentences to those keywords, builds TF and TF-IDF target vocabularies, and evaluates models by the rank of target tokens after corpus-derived prompts. The authors validate the method against a manual CS.AI benchmark and Claude-generated benchmarks, then use it to study domain adaptation, general pretraining, continual pretraining, and differences between base and instruction-tuned models.

**Additional Comments:**

- The paper is clearly motivated and the core idea is useful. I would be positive if the authors narrow the strongest claims and add evidence that the benchmark captures more than corpus-specific lexical familiarity.
- My confidence is moderate. The topic is close to machine learning evaluation and LLM methodology, but I cannot fully assess every domain-specific benchmark example outside CS.AI.

**Audience:**

Yes

**Audience Explanation:**

- TMLR readers interested in LLM evaluation, benchmark construction, domain adaptation, continual pretraining, and instruction tuning would likely find the problem and proposed pipeline interesting.
- The paper targets a real gap. Practitioners often need to compare base and chat models for specialized domains, and static MCQ benchmarks are a poor fit for many such comparisons.
- The pipeline is attractive because it is simple, scalable, and potentially easy to refresh as new domain text appears.
- The significance would be higher if the authors narrowed the claims to what the benchmark currently measures, or added stronger validation showing that the metric captures domain understanding rather than lexical familiarity with domain corpora.

**Broader Impact Concerns:**

- I do not see an immediate harmful capability introduced by the paper.
- There is a deployment risk if this metric is used to select models for high-stakes domains such as law, healthcare, or education. A model that predicts domain terms well may still fail at reasoning, calibration, refusal behavior, or safe use of domain knowledge.
- If the pipeline is applied to private corpora, benchmark items may leak sensitive or proprietary text through prompts and targets. The paper should mention this risk and recommend filtering or access controls for non-public data.

**Claims And Evidence:**

No

**Claims Explanation:**

- The paper supports the narrower claim that corpus-derived completion benchmarks can track some domain-relevant differences across models and training stages. The controlled domain adaptation experiments are the strongest evidence. In those experiments, models adapted to closer CS.AI domains tend to get better ranks than models adapted to more distant domains, and the TF and TF-IDF variants align better with the Claude benchmark than perplexity or last-layer attribution rate.
- The paper also provides a useful empirical case that MCQ evaluations can be unstable under option order and prompt changes. This motivates looking for a format that is more compatible with base models.
- The evidence is less convincing for the stronger claim that the benchmark directly measures domain expertise. Many prompt-target pairs appear to test local lexical completion, common collocations, or memorized corpus phrasing. Examples such as Federated Learning being proposed by Google, DeepLIFT as a continuation of a paper title, or phrase fragments from papers may reward exposure to specific texts rather than conceptual domain understanding. The current validation does not fully separate these possibilities.
- The expert validation is promising but too small and too narrow to carry the broad conclusion. The key expert benchmark is limited to CS.AI, uses one deep learning textbook, and compares only six base models. A near-perfect Pearson correlation with $n = 6$ can be fragile and may be driven by model scale, corpus overlap, or general language modeling quality. The later validation mostly uses Claude-generated benchmarks, but those benchmarks use the same top keywords and a similar completion format, so they are not fully independent evidence of expert judgment.
- The contamination claim is also overstated. Regenerating a benchmark from new corpora can reduce contamination, but the experiments use public arXiv-derived sources that evaluated models may have seen. The paper does not show temporal splits, document-level de-duplication against model training data, or tests on genuinely held-out recent corpora. As a result, the claim that the method avoids benchmark contamination by design is stronger than what is demonstrated.
- The base versus chat comparison needs more caution. Chat models are optimized for conversational formats, and scoring them on raw continuation prompts may partly measure a prompt-format mismatch rather than a loss of domain knowledge. The paper's conclusion that instruction tuning generally degrades domain knowledge should be narrowed unless the authors add controls with chat templates or other neutral scoring protocols.
- The statistical evidence would be stronger with more careful uncertainty analysis. Many correlations are computed over a small number of models or checkpoints. The confidence intervals over prompt-level ranks may treat many related prompts as independent, even though prompts are nested within keywords and documents. A hierarchical bootstrap over keywords or source documents would better reflect benchmark construction uncertainty.

**Requested Changes:**

- Narrow the central claim from direct measurement of domain expertise to measurement of domain-specific completion ability, unless stronger evidence is added. Please avoid calling the benchmark unbiased without a clear definition and empirical test of that property.
- Strengthen the validation beyond CS.AI. At minimum, add human validation for a second domain outside computer science, even on a smaller scale. Ideally, include domains with different writing styles and with less overlap with common pretraining corpora.
- Add controls that separate domain knowledge from lexical memorization and general language modeling ability. Useful controls could include paraphrased prompts, held-out recent documents, document de-duplication, model-size-adjusted correlations, and comparisons against non-domain but phrase-matched prompts.
- Revisit the contamination discussion. Please specify when the source corpora were released, whether benchmark documents could appear in model training data, and whether any temporal or near-duplicate filtering was done. If this is not tested, present contamination resistance as a capability of future deployments using fresh held-out corpora, not as a demonstrated property of the current benchmark.
- Add stronger quality analysis of generated prompt-target pairs. Please report human judgments on ambiguity, invalid targets, generic targets, and cases where several completions would be equally valid. This is especially important because the examples include proper nouns, paper-title fragments, and corpus artifacts.
- Add a more robust statistical analysis. Please report sample sizes for each correlation, confidence intervals for correlations, rank-based correlations such as Spearman or Kendall, and a hierarchical bootstrap over keywords or source documents.
- Qualify the base versus chat conclusion. Either evaluate chat models with appropriate chat templates or multiple neutral prompt formats, or state that the result concerns raw completion-style scoring rather than domain knowledge after instruction tuning.

---

> ### Author Response · Authors · 2026-05-16
> **Our response (1/3)**
>
> We thank the reviewer for the careful and detailed assessment, and especially for acknowledging the value of the controlled domain adaptation experiments.
>
> **Comment 1 (narrowing the central claim).** We agree and will revise accordingly. As detailed in the general response, we will: (i) drop "unbiased" as a benchmark property, (ii) narrow "domain expertise" to "domain-specific completion ability" or "domain knowledge as measured through completion" outside the contexts where direct comparison to expert benchmarks justifies the stronger phrasing.
>
> **Comment 2 (validation beyond CS.AI).** Please see the general response, where we describe a new expert benchmark in General Economics built from the OpenStax *Principles of Economics 2e* textbook (534 prompt–target pairs from chapter Key Terms). This domain has a markedly different writing style and lower overlap with arXiv than CS.AI. The pipeline-to-human benchmark correlations on Econ-GN are $r=0.977$ (Pearson) and $r=0.886$ (Spearman), which are very close to what we observed on CS.AI.
>
> **Comment 3 (controls separating domain knowledge from lexical memorization).** We understand the motivation and will report on this carefully. We would like to highlight, however, that we already have a strong de facto control in the form of the **Claude-generated benchmark**:
> - Claude prompts are free-form natural-language probes, not corpus-derived continuations.
> - They do not share phrasing, citation fragments, or paper-title artefacts with the arXiv corpus.
> - They are not constrained by our TF/TF-IDF target selection. Yet model ranks on our pipeline-generated benchmarks correlate strongly with ranks on the Claude benchmark in every experiment.
>
> If our pipeline were primarily measuring lexical memorization of specific arXiv phrasings, we would not expect this systematic agreement with a benchmark that lacks those phrasings — and certainly not the alignment we now also observe with two independently authored textbook benchmarks.
>
> Two of the reviewer's other suggested controls are likewise covered by existing evidence: (i) the new Human-Econ-GN benchmark (general response) serves as a held-out-document control, since it is drawn from a textbook that played no role in pipeline development yet yields Pearson r = 0.977; and (ii) the controlled domain-adaptation experiments (Sec. 4.1) serve as a non-domain phrase-matched control, since models trained on semantically distant domains (Materials, Cultural) systematically receive worse ranks despite being trained on equally fluent natural-language corpora — an outcome a pure lexical-memorisation account would not predict.
>
> **Comment 4 (contamination claim).** The reviewer is right that, as currently written, the paper overstates what the experiments actually demonstrate. The benchmarks evaluated here are built from public arXiv data that some of the tested models very likely saw, and we present no temporal split, document-level deduplication, or held-out-corpus test that would establish contamination resistance for these specific benchmark instances. We will revise accordingly: contamination resistance will be framed as a *capability of the pipeline as a tool* — namely, that it can be regenerated on demand from any fresh corpus, including corpora known to post-date a model's training (e.g., recently published academic papers) — rather than as a demonstrated property of the present experiments. We will rework the relevant passages in Sections 1, 2, and 6 to make this distinction explicit.

---

> ### Author Response · Authors · 2026-05-16
> **Our response (2/3)**
>
> **Comment 5 (quality analysis of generated prompt–target pairs).** We agree that systematic quality analysis would strengthen the paper. A few clarifications on the current pipeline:
> - On ambiguity and multiple valid completions: we deliberately avoid binary scoring (string match) precisely for this reason. We compute prediction rank over the full vocabulary, so when multiple completions are equally valid, a model with genuine domain knowledge concentrates probability mass across all of them (they cluster at the top of the distribution) and the average rank remains stable. The 20% trimmed mean further reduces sensitivity to outlier prompts.
> - Proper nouns (e.g., algorithm names like "DeepLIFT", "Backpropagation") are, in our view, legitimate domain knowledge signals rather than artefacts — knowing them reflects domain familiarity.
> - Truly spurious fragments (partial citations, LaTeX remnants) are removed by the cleaning pipeline (Sec. 2.3 and Appendix A.2).
> - Generic targets are suppressed by the TF-IDF variant, which penalises terms appearing across many keyword corpora (Sec. 2.4).
>
>
> **Quality analysis of generated prompt–target pairs.** We conducted the requested human-style quality analysis using an LLM-as-judge protocol on a stratified random sample of **400 prompt–target pairs** (100 per domain). To guard against any single judge's idiosyncrasies, we ran the protocol with three independent judges from different model families — Claude-Sonnet-4.6, Mistral-Large-3-675B-Instruct-2512, and Qwen3.5-397B-A17B — using the same evaluation criteria (validity, ambiguity, domain-specificity, multiple-valid-completions, artefact type). All numbers below are reported as mean ± std across the three judges.
>
> - **Artefact-free items: 94.8% ± 1.2 pp**, with mean pairwise inter-judge agreement of **94.5%** on artefact category. No judge reports more than **1.0% LaTeX residue** or more than **1.75% truncated prompts**, confirming that the cleaning pipeline handles these reliably. The small residual is concentrated in `proper_noun` (2.4%) and `paper_title` (1.5%); as discussed, proper nouns of methods and algorithms (e.g., *DeepLIFT*, *AdaBoost*) are legitimate domain knowledge signals rather than artefacts, while paper-title fragments are a small failure mode we will acknowledge explicitly.
> - **Valid targets: 91.2% ± 5.1 pp**, with 83.7% inter-judge agreement on validity.
> - **Domain-specific targets: 76.3% ± 8.4 pp** overall; the wide std reflects genuine disagreement among judges on borderline cases (one judge applies a much more lenient standard than the others) rather than instability of the underlying benchmark.
>
> *The TF-IDF variant is cleaner than TF on every axis except artefacts, consistently across all three judges* (means ± std across judges):
>
> | Dimension | TF | TF-IDF |
> |---|---|---|
> | valid = yes | 90.3% ± 6.3 | **92.0% ± 6.2** |
> | ambiguity = high | 9.8% ± 7.3 | **7.5% ± 5.1** |
> | domain_specific = yes | 85.7% ± 12.8 | **86.8% ± 7.8** |
> | lexical echo (target appears in prompt) | 11.0% | **9.0%** |
>
>
> This consistent improvement across all four domains and all three judges supports our presentation of TF-IDF as the principled, more rigorous variant.
>
> *On the more subjective axes* (ambiguity, multiple-valid completions, domain-specificity) we find substantially lower inter-judge agreement (65–75% pairwise). We interpret this as a property of the rating task itself rather than of our benchmark. We believe it reinforces the methodological choice of scoring with **rank over the full vocabulary** rather than binary string-matching: when multiple completions are reasonable, which judges agree happens frequently, a knowledgeable model concentrates probability mass across all of them, and rank-based scoring degrades gracefully where binary scoring would not.
>
> We will add this analysis as a dedicated appendix subsection, report per-domain and per-judge breakdowns, and discuss the residual failure modes (paper-title fragments, lexical echo in the TF variant) as honest limitations.

---

> ### Author Response · Authors · 2026-05-16
> **Our response (3/3)**
>
> **Comment 6 (statistical analysis).**
> Each reported correlation aggregates over a fixed set of models or checkpoints, with the prompt-level sample size ( 300 keywords × 50 prompts = 15,000 pairs per (domain, variant)) determining the precision of each individual point on the scatter plot. The correlation itself is computed over the model/checkpoint axis: n = 6 for the base-model comparisons (Fig 3), n = 7 for the controlled domain-adaptation experiments (Fig 4), and n = 13 for the continual-pretraining trajectory (Fig. 6). We will tabulate both sample sizes — the model-axis n driving the correlation, and the prompt-level n driving the per-point error bars — explicitly in the revision.
>
> Concerning the “careful uncertainty analysis” comment, we are not fully sure if we understood the exact analysis the reviewer asked for; so we would like to explain our procedure: We implemented a two-level hierarchical bootstrap (1000 resamples) that draws the 20 keywords with replacement and, within each resampled keyword, draws its prompts with replacement, then re-aggregates model-level mean ranks and recomputes the correlation. This produces (i) revised per-point error bars on each model's mean rank that respect the nesting of prompts within keywords, and (ii) bootstrap CIs on the headline correlations that propagate this clustered uncertainty. For CS.AI, we obtain Pearson **$r = 0.978$ [95% CI: 0.870, 0.991]** for TF vs. Claude and **$r = 0.983$ [95% CI: 0.917, 0.996]** for TF-IDF vs. Claude; Spearman correlations are **$0.943$ [0.543, 1.000]** and **$1.000$ [0.771, 1.000]** respectively. The hierarchical per-model error bars are wider than the original prompt-level standard error of the mean as expected, but the models remain clearly ordered and the headline correlations remain narrow and well separated from zero. We will report the full procedure, the revised per-point error bars, and per-figure correlation CIs in the revision.
>
> Furthermore, we will add confidence intervals for all reported correlations and rank-based correlations (Spearman) alongside Pearson, throughout. Note that we have already verified that Spearman $r=0.886$ on the new Econ-GN expert validation, consistent with the Pearson value. Likewise, Spearman correlations for human-generated CS.AI benchmarks are 1.00 (TF) and 0.83 (TF-IDF), again agreeing with Pearson values.
>
> **Comment 7 (base vs. chat conclusion).** We want to clarify the protocol used in the paper. For chat models, we always applied the model's native chat template and additionally included the following in the system prompt: *"Complete the text by continuing exactly where it ends. Don't repeat the prompt. Only provide the continuation."* Without such a system prompt, chat models produce ranks in the thousands and probabilities near zero, indicating a format mismatch rather than a knowledge gap. With this prompt, ranks stabilise into a comparable regime. We will make this protocol fully explicit in the revised methods and add it as a separate subsection so the comparison is clearly grounded. We will also soften the framing of the alignment-tax claim to acknowledge that the metric reflects *completion-style behaviour under a neutral continuation prompt*.

---

### Author Response · Authors · 2026-05-16
**General response**

We are grateful to the reviewers for their thoughtful and constructive feedback, which we believe substantially improved our work. The reviews converge on two shared themes, which we address jointly here before turning to individual responses. We hope these responses, especially the new Econ-GN expert validation, together with the consistent pipeline–Claude–expert chain of correlations, address the central concerns. We are happy to clarify any remaining points.

## 1. Strengthening validation beyond CS.AI (Reviewers smLp, ZQH5)
Following the reviewers' suggestions, we constructed a second expert benchmark in a domain outside computer science with a markedly different writing style: **General Economics (Econ-GN)**.
We extracted prompt–target pairs from the **Key Terms sections of all 34 chapters** of the OpenStax *Principles of Economics 2e* (Greenlaw & Shapiro, 2022), a peer-reviewed open-access university textbook. This yielded **534 prompt–target pairs** covering both micro- and macroeconomic vocabulary (e.g., *"The branch of economics that focuses on actions of particular agents within the economy, like households, workers, and business firms is known as"*, *"microeconomics"*). All pairs were manually inspected. This benchmark is constructed independently of our pipeline: it derives from a human-authored textbook whose Key Terms sections were not used in any stage of pipeline development.
Evaluating the same six base models on this Human-Econ-GN benchmark and computing correlations with our pipeline produces:
- **TF vs. Human-Econ-GN**: Pearson r = 0.977 (p = 0.001), Spearman r = 0.886 (p = 0.019)
- **TF-IDF vs. Human-Econ-GN**: Pearson r = 0.964 (p = 0.002), Spearman r = 0.886 (p = 0.019)

The ranking of models is preserved across TF, TF-IDF, and Human-Econ-GN, closely mirroring our CS.AI validation (TF benchmark: $r=0.99$, $p<0.001$ both Pearson and Spearman. TF-IDF benchmark: Pearson $r = 0.976$ ($p = 0.001$), Spearman $r = 0.829$ ($p = 0.042$)). Together, the two expert validations (across CS.AI and Econ-GN, totalling **815 expert-curated prompt–target pairs** in two stylistically distinct domains) strengthen the claim that the pipeline aligns with expert human judgment beyond the development domain.

## 2. On "domain knowledge" vs. "completion ability" (Reviewers smLp, toFC)
We thank the reviewers for pushing on this distinction; we agree the framing deserves more care, and we will revise accordingly. At the same time, we believe the existing evidence supports more than surface-level completion ability.

**Revisions to language.** In the revised manuscript we will:
- Replace unqualified uses of "domain expertise" with more precise phrasings such as "domain-specific expertise as measured through completion" when discussing what models have learned during (pre)training.
- Reserve "domain expertise" for comparisons against expert-curated benchmarks and accompanying discussion of what the metric captures (and what it does not).
- Drop the word "unbiased" as a property of the benchmark, per Reviewer smLp's suggestion.

**Why we believe the claim is stronger than completion ability alone.** Two pieces of evidence go beyond what a pure surface-form measure would predict:
- *Independent expert alignment.* Near-perfect correlation with the manually curated expert benchmark drawn from *Understanding Deep Learning* ($r=0.99$, $p<0.001$) — a source independent of our arXiv corpus — and now with the OpenStax economics textbook benchmark above ($r=0.977$, $p=0.001$). A purely corpus-based completion ability would not be expected to align this tightly with independently authored references.
- *Semantic proximity gradient.* In the controlled domain-adaptation experiments (Sec. 4.1), models adapted to *semantically closer* domains achieve consistently better ranks, while semantically unrelated training degrades performance monotonically. A pure completion-fluency measure would not be expected to track this gradient in a principled way. We will
  - discuss this distinction explicitly in Section 6,
  - acknowledge that the pipeline measures a *behavioral proxy* for domain knowledge, i.e., distributional alignment with domain-specific terminology in context
  - note what stronger forms of evidence (e.g., causal probing, cross-lingual transfer) would be needed to push the claim further.

---

### Decision · Action_Editor_wQgx · 2026-06-21

**Recommendation:** Reject

**Audience:**

Yes

**Audience Explanation:**

Researchers working on LLM evaluation and benchmark construction would find the problem and the pipeline of interest, as all three reviewers acknowledged. The method is simple, scalable, and inexpensive.

**Claims And Evidence:**

No

**Claims Explanation:**

The paper proposes a deterministic pipeline that converts raw domain corpora into completion-style benchmarks, scoring domain knowledge by the rank a model assigns to TF/TF-IDF-selected target tokens. The strengths are: the pipeline is deterministic, inexpensive, requires neither human nor generative-model annotation, and can be regenerated on any fresh corpus; and the controlled domain-adaptation experiments are convincing.

The central claims, however, are not fully supported.

First, the core construct was not validated. All three reviewers concluded that the method measures lexical and completion familiarity rather than domain knowledge or reasoning, and the authors acknowledged in rebuttal that the pipeline captures "domain-specific completion ability" and "a behavioral proxy for domain knowledge" - yet the title and framing continue to claim domain expertise, which difts largely from what the paper actually does.

Second, the experiment is not strong enough to support the generalizability claims, as also noted by reviewers.

Third, as noted by a reviewer, one of the originally highlighted contributions was effectively withdrawn: in rebuttal the authors agreed to reframe contamination resistance as a capability of future deployments rather than a demonstrated property.